# Recent Link Classification on Temporal Graphs Using Graph Profiler

**Muberra Ozmen** [*]                                                                      *muberra@squareup.com*
*CashApp*
*Montreal, QC, Canada*

**Thomas Markovich** [†]                                                                   *tmarkovich@squareup.com*
*CashApp*
*Cambridge, MA, USA*

**Reviewed on OpenReview:** *https://openreview.net/forum?id=BTgHhOgSSc&referrerD*

## Abstract

The performance of Temporal Graph Learning (TGL) methods are typically evaluated on the *future link prediction* task, i.e., whether two nodes will get connected and *dynamic node classification* task, i.e., whether a node's class will change. Comparatively, *recent link classification*, i.e., to what class an emerging edge belongs to, is investigated much less even though it exists in many industrial settings. In this work, we first formalize recent link classification on temporal graphs as a benchmark downstream task and introduce corresponding benchmark datasets. Secondly, we evaluate the performance of state-of-the-art methods with a statistically meaningful metric *Matthews Correlation Coefficient*, which is more robust to imbalanced datasets, in addition to the commonly used average precision and area under the curve. We propose several design principles for tailoring models to specific requirements of the task and the dataset including modifications on message aggregation schema, readout layer and time encoding strategy which obtain significant improvement on benchmark datasets. Finally, we propose an architecture that we call *Graph Profiler*, which is capable of encoding previous events' class information on source and destination nodes. The experiments show that our proposed model achieves an improved Matthews Correlation Coefficient on most cases under interest. We believe the introduction of recent link classification as a benchmark task for temporal graph learning will be useful for the evaluation of prospective methods within the field.

## 1 Introduction

Graphs provide convenient structures to represent interactions or relationships between entities by modeling them as edges between vertices. Using this representation allows one to build models that capture the interconnected nature of complex systems such as social networks (El-Kishky et al., 2022; Wu et al., 2022; Gao et al., 2021) or transaction graphs (Liu et al., 2020; Zhang et al., 2022). *Graph Representation Learning (GRL)* rose in popularity due to the desire to apply deep learning to graph structured problems (Zhou et al., 2020; Wu et al., 2020; Hamilton, 2020). Indeed, GRL has provided significant advances in fraud detection (Liu et al., 2020; Zhang et al., 2022), recommendation systems (Wu et al., 2022; Gao et al., 2021), chemistry and materials science (Pezzicoli et al., 2022; Reiser et al., 2022; Bongini et al., 2021; Han et al., 2021; Xiong et al., 2021), traffic modeling (Rusek et al., 2019; Chen et al., 2022), and weather simulation (Keisler, 2022; Ma et al., 2022), among other possible applications. Many of these graph machine learning tasks can be understood as either link prediction (Chen et al., 2020; Cai et al., 2019; Zeb et al., 2022; Chamberlain et al.,

---

[*]Also attending McGill University during this research,
[†]Corresponding author

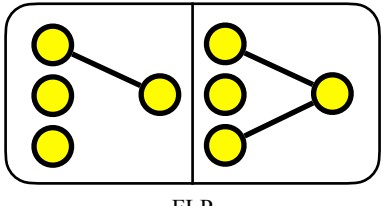 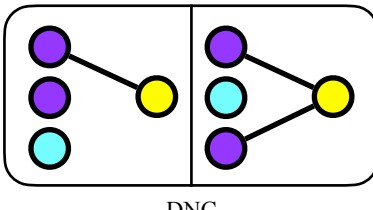 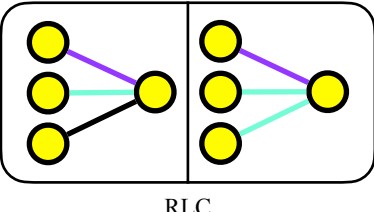

| FLP | DNC | RLC |
|---|---|---|
| Which nodes will be connected next? | What will be the node colour in the future? | What will be the colour of a new edge? |

Figure 1: Differences between TGL tasks. **Future Link Prediction (FLP):** Given a temporal graph and a pair of nodes observed at a timestamp, a function is learned which predicts the probability of these two nodes linking at a later timestamp. **Dynamic Node Classification (DNC):** Let blue and purple colours represent the node classes which change over time. Given the current node classes along with the set of edges at a time point, the aim is predicting the node classes at a later timestamp. **Recent Link Classification (RLC):** Let the colours blue and purple represent the edge classes, which are observed some time after the emergence of a link. That is, at the time when a link is first detected, the link class is unknown. Black edge have been recently observed but the class is not yet observed. In this setting, the aim is to predict recent link classes given a history of previous interactions.

2022) or node classification tasks (Kipf & Welling, 2016; Zhang et al., 2019). Much of the early work focused on the scenario where the graph is static. Acknowledging that many tasks in industrial settings involve graphs that evolve in time, researchers defined a sub-problem of GRL called *Temporal Graph Learning (TGL)*, with time dependent versions of the original static tasks, yielding *Future Link Prediction (FLP)* and *Dynamic Node Classification (DNC)* respectively (Kumar et al., 2019; Arnoux et al., 2017). The former task, FLP, seeks to predict whether two vertices will be connected at some specified future time; while the latter, DNC, seeks to predict the class of a vertex at a future time (See Figure 1). Both of these tasks have a variety of applications in real world, e.g., predicting the probability of two people forming a tie or of a person deactivating their account on social media platforms, corresponding to FLP and DNC tasks respectively (Min et al., 2021; Song et al., 2021; Frasca et al., 2020; Zhang et al., 2021a).

This begs the question – "In analogy to the dynamic node classification task, is there a temporal link classification task we can define?" And indeed, there is a third common task that is based on static link classification (Wang et al., 2023), which we term *recent link classification (RLC)*. RLC is present in industrial settings but has yet to be formalized in a research setting. The task of RLC requires that one classify a link that has been observed to exist but does not as yet have a known label. This task is important for settings where we wish to classify the interaction but labels may be delayed. In social media settings, this could involve classifying interactions as abusive, and in a transaction network it could involve classifying a transaction as potentially fraudulent. More formally, we define RLC as a task in which the predictive algorithm is given the source and destination entities of a recent interaction and its features, e.g., textual content of the post; and it must predict the target class value. The task of classifying recent interactions has typically been treated as a tabular data task, arising in applications such as fraud detection (Sarkar, 2022). This approach represents each interaction between two entities as a sample with associated features. The sample can thus be represented as a row of a table, where each column is a feature. There is usually an (implicit) assumption that the samples are independent. This neglects the fact that entities may participate in multiple interactions and thus induce dependencies between the samples. In practice, one option to address this deficiency is feature engineering. For tabular tasks, it has been previously observed that features representing counts, such as how many times one user has previously liked another user's posts, provide significant metric uplift (Wu et al., 2019; Shan et al., 2016). While the incorporation of such features is not explicitly graph machine learning, we take the view that these features are the result of manual feature engineering inspired by graph-based intuition (Zhang et al., 2021b; Martínez et al., 2016; Chamberlain et al., 2022). Therefore, we believe that explicitly formulating RLC as a graph learning task and highlighting suitable datasets for the temporal graph machine learning community will encourage progress towards better solutions for an industrially relevant problem.

With this motivation in hand, we formalize two research questions that we wish to answer in this work: "**Q1: How does recent link classification differ from future link prediction?**" and "**Q2: What are the most critical design principles of recent link classification?**". We answer the first question through a comparative study of baseline methods on both tasks. We answer the second question by exploring a variety of model building blocks, some published previously, and some novel. In answering these research questions, we contribute a new TGL task, a new figure of merit, a measure of edge-homophily, and a non-exhaustive set of design principles that comprise a design space for this new machine learning task.

## 2  Related Work

There exist two main models for dynamic graphs: discrete-time dynamic graphs and continuous-time dynamic graphs (Zhou et al., 2022). Discrete-time dynamic graphs manifest as sequences of static graph snapshots captured at distinct time intervals. Continuous-time dynamic graphs, on the other hand, offer greater generality and can be expressed as timed sequences of events. These events may involve the addition or deletion of edges, addition or removal of nodes, and transformations of node or edge features. Traditional static graph representations fall short in capturing the inherent temporal dynamics. TGL seeks to address this limitation by extending the principles of graph-based models to time-varying structures, enabling a comprehensive understanding of how relationships unfold and transform over different time intervals. Gao & Ribeiro (2022) develop a framework to analyze TGL architectures, and categorize methods in the literature into two groups: 'time-and-graph' and 'time-then-graph'. Time-and-graph based architectures learn the evolution of node representations by building a representation for each graph snapshot. DySAT (Sankar et al., 2020) and Evolve GCN (Pareja et al., 2020) fall under this category. Time-then-graph based architectures, on the other hand, construct a multi-graph using the memory of all past observations and build a static graph to learn the node representations. Most well-known methods which fall under this category are TGN (Rossi et al., 2020) and TGAT (da Xu et al., 2020). Our research aligns with the 'time-then-graph' approach. To that end, we provide a brief review of the methods that we have found influential.

TGN is a message passing based encoder which learns graph node embeddings on a continuous-time dynamic multi-graph represented as a sequence of time-stamped events. It involves three main building blocks: (1) message function, (2) memory function and (3) embeddings module. At each event, a message function in the form of event-adopted MLPs, calculates aggregated messages to pass on parties involved (i.e., nodes). A memory function, such as an LSTM or GRU, updates the memory state of each node by the aggregated messages. An embedding module calculates the new state of node embeddings as a function of memory states and events. TGAT incorporates the self-attention mechanism as a foundational element and introduce a time encoding approach which allows the network to treat node embeddings as temporal functions, thereby enabling it to predictively generate embeddings for both newly introduced and previously existing nodes as the graph undergoes changes over time. In their work, Wang et al. (2021) present CAWN, a method for the inductive representation of temporal networks which utilize temporal random walks. Temporal random walks enable to encapsulate network dynamics by implicitly extracting temporal network motifs, thereby circumventing the intensive process of motif identification and enumeration. The methodology incorporates a distinctive anonymization technique wherein node identities are obscured by their visitation frequencies within a series of sampled walks which preserve the inductive nature of the model. Yu et al. (2023) introduce a transformer-based framework designed for dynamic graph learning. The proposed architecture focuses on learning from the historical first-hop interactions between nodes. The learning process is facilitated by two key strategies: a neighbor co-occurrence encoding scheme that uncovers the relationships between the source and destination nodes through their historical interactions; and a patching method that segments each interaction sequence into smaller patches. The Graph Mixer (Cong et al., 2023) takes a simpler view by constructing a model that has a fixed time encoding, alongside a node encoder and a link encoder; the encodings are used as inputs for a link classifier that is trained to predict the existence of a link. All state-of-the-art methods experiment on benchmark FLP and DNC datasets. Despite its simple infrastructure, Graph Mixer is able to achieve state-of-the-art performance on both FLP and DNC tasks.

In general, TGL models are developed to address FLP and DNC tasks. However, the performance is highly sensitive to non-architectural hyperparameters such as batch size and the negative sampling strategy. These hyperparameters are difficult to tune and often represent a trade-off between computational efficiency and

accuracy, so optimizing them can provide a distorted view of expected performance in industrial settings. Additionally, to the best of our knowledge, there is no method that is capable of generating embeddings for the relatively common setting that event labels are obtained with delay. We address these issues by formulating RLC as a new benchmark task for TGL and developing Graph Profiler which enables the construction of dynamic node profiles by taking the feature and label information associated with previous interactions into account. Graph Profiler is capable of maintaining a long-term view of an entity's profile that can capture long-term preferences.

## 3 Problem Statement

A graph, $\mathcal{G}$, is composed of a set of vertices, $\mathcal{V}$, and edges $\mathcal{E}$, where each edge, $i \to j$, indicates an interaction between a pair of vertices $i$ and $j$. In most cases, the graph is constructed with entities as the vertices and interactions between those entities as edges. In the case of social networks vertices might be users and their posts, and edges might involve interactions either between users, such as follows or blocks, or interactions between users and posts such as likes or comments. For a general RLC task we are given a set of source entities and destination entities, $\mathcal{V}_{\mathrm{src}}$ and $\mathcal{V}_{\mathrm{dst}}$, and a set of interactions between them $\mathcal{E} = \{(s_i, d_i, t_i, \mathbf{x}_i, \mathbf{y}_i)\}_{i=1}^{M}$; such that the interaction from source entity $s_i \in \mathcal{V}_{\mathrm{src}}$ to destination entity $d_i \in \mathcal{V}_{\mathrm{dst}}$ is realized at time $t_i$ and associated with a raw feature vector of $\mathbf{x}_i \in \mathbb{R}^{d_{\mathrm{msg}}}$ where $d_{\mathrm{msg}}$ denotes the number of features. We consider that interactions are effectively instantaneous, or that the timestamp marks the completion of the interaction. We consider settings where there are $m$ classes (or types) of interaction. Each interaction is thus associated with a ground-truth target class, represented as a binary vector $\mathbf{y}_i = (y_{i,1}, \ldots, y_{i,m})$ such that $y_{i,j} = 1$ if interaction $i$ belongs to the $j^{\mathrm{th}}$ class and $y_{i,j} = 0$ otherwise. The aim is to learn a classifier that maps features, timing, source and destination entities of an interaction to a class, given access to a history of preceding interactions. Given a new interaction from source $s \in \mathcal{V}_{\mathrm{src}}$ to destination $d \in \mathcal{V}_{\mathrm{dst}}$ with features $\mathbf{x} \in \mathbb{R}^{d_0}$ which is realized at time $t$, let $\mathcal{E}_{<t} = \{(s_i, d_i, t_i, \mathbf{x}_i, \mathbf{y}_i) \in \mathcal{E} : t_i < t\}$ denote the preceding observations, and let $\hat{\mathbf{y}} = (\hat{y}_1, \ldots, \hat{y}_m)$ denote the predicted target class likelihoods by the classifier, i.e., $f(\mathbf{x}, (s, d, t), \mathcal{E}_{<t}) = \hat{\mathbf{y}}$. Traditionally, the quality of estimation is evaluated by the cross entropy loss $\mathbb{L}_{\mathrm{ce}}(\mathbf{y}, \hat{\mathbf{y}}) = -\sum_{j=1}^{m} y_j \log(\hat{y}_j)$ during training. In case that $\mathcal{G}$ is not bipartite, i.e., $\mathcal{V}_{\mathrm{src}} \cap \mathcal{V}_{\mathrm{dst}} \neq \emptyset$, this formulation requires us to learn different representations for a vertex as the sender and the receiver of a message. In the basic problem formulation, we assume that vertices are not attributed, i.e., raw features are not observed for them. At the time of the event occurrence, we observe the identities of the source and destination, as well as the raw edge features. None of these changes over time. The observation of the edge label is delayed.

We identify three other subfields of GRL that are potentially relevant to the problem setting formulated by RLC:

**Dynamic Graph Anomaly Detection:** Anomaly detection within dynamic graphs focuses on identifying trends that starkly contrast with the norm over time (Yu et al., 2018; Zheng et al., 2019; Behrouz & Seltzer, 2022). Specifically, in scenarios where there are only two types of edge labels and a pronounced class imbalance exists, this approach to anomaly detection proves pertinent to RLC task. However, its applicability does not extend universally across all other instances.

**Link Sign Prediction:** Among static graph learning approaches, the research most pertinent to RLC predominantly focus on link sign prediction. Link sign prediction aims to determine the positive or negative nature of relationships in networks (Song & Meyer, 2015; Aggarwal et al., 2016; Dang & Ignat, 2018; Chen et al., 2023). Methods in this domain often leverage structural balance theories and employ various machine learning techniques to predict the sign of unlabeled edges based on known relationships.

**Dynamic Link Property Prediction:** Huang et al. (2023) extends the definition of FLP to dynamic link property prediction by enabling it to predict aspects of a link beyond mere existence. However, their work lacks an experimental framework that explores edge properties other than existence.

## 4 Graph Profiler

In this section, we introduce *Graph Profiler*, a simple architecture that is designed to learn entity profiles, or time-aggregated representations, by processing previously observed interactions. Subsequently it uses the

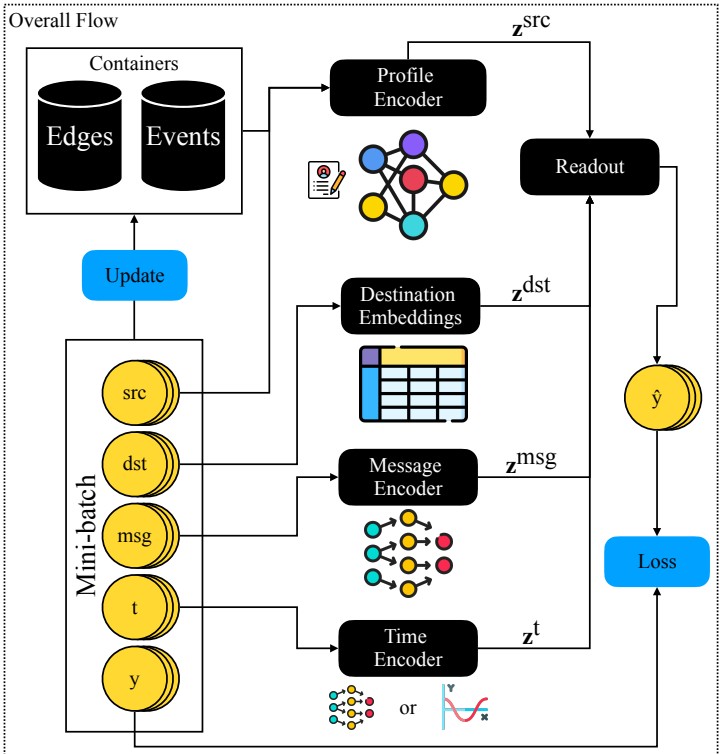
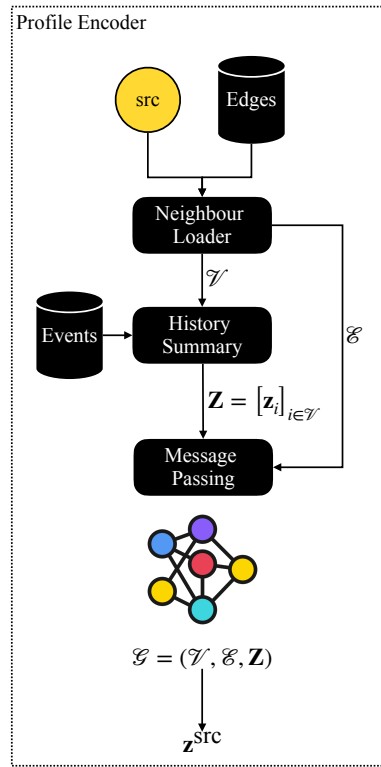

Figure 2: Overview of Graph Builder. **Left:** Overall flow. Given a mini-batch of interactions, where each interaction is from a source node `src` to a destination node `dst`, observed at time `t`, associated with features `msg` and belongs to class `y`, Graph Profiler follows these steps: (1) The profile encoder calculates source node embeddings based on the history of `src` (i.e., previous edges and events that are associated with `src`). (2) The identity of `dst` is used to retrieve the embedding of the destination node from the table of destination embeddings. (3) The message encoder performs a linear transformation on the edge features. (4) The time encoder projects timestamp `t` into a time embedding. (5) The readout layer combines source $\mathbf{z}^{\mathrm{src}}$, destination $\mathbf{z}^{\mathrm{dst}}$, message $\mathbf{z}^{\mathrm{msg}}$ and time $\mathbf{z}^{\mathrm{t}}$ embeddings to predict the class of the interaction. Afterwards, the containers are updated with the information in the mini-batch. **Right:** Profile builder. Given the current set of edges, the neighbourhood of `src` is loaded. The past events are used to summarize the history of all nodes that belong to the neighbourhood of `src`. The node embeddings on the ego graph of `src` are updated using message passing to obtain the final source node embedding $\mathbf{z}^{\mathrm{src}}$.

constructed entity profiles, together with observed interaction features, to make classification decisions about recent interactions. Graph Profiler is composed of five main learnable modules; **profile encoder** $f_{\mathrm{profile}}(\cdot)$, **message encoder** $f_{\mathrm{msg}}(\cdot)$, **destination encoder** $f_{\mathrm{dst}}(\cdot)$, **time encoder** $f_{\mathrm{time}}(\cdot)$, and **readout** $f_{\mathrm{rlc}}(\cdot)$, and two containers at time $t$; previous **events** $\mathcal{H}_t$ and meta-path **edges** $\mathcal{M}_t$. For a given set of interactions (edges), $\mathcal{E} = \{(s_i, d_i, t_i, \mathbf{x}_i, \mathbf{y}_i)\}_{i=1}^{M}$, the event and meta-path containers at time $t$ are defined as follows:

$$\mathcal{H}_t = \{i \in \{1, \ldots, M\} : t_i < t\}, \tag{1}$$

$$\mathcal{M}_t = \{(u, v) : \exists i, j \in \mathcal{H}_t; (s_i = u) \wedge (s_j = v) \wedge (d_i = d_j)\}. \tag{2}$$

The event container simply stores the indices of the interactions observed up to time $t$. The meta-path container stores the pairs of source nodes that have interacted with the same destination node at least once prior to time $t$ (See Figure 3). Graph Profiler proceeds in the following way to perform training. We denote by $\mathcal{E}_{\mathrm{batch}} \subset \mathcal{E}$ the mini-batch of interactions that is being processed. Let $t_{\mathrm{current}} = \min_{j \in \mathcal{E}_{\mathrm{batch}}} t_j$ denote the first interaction time in the batch, and $d_{\mathrm{model}}$ denote the dimensionality of the embeddings constructed by the model. Given an interaction $(s_i, d_i, t_i, \mathbf{x}_i) \in \mathcal{E}_{\mathrm{batch}}$, the computation proceeds through each module as follows:

1. The profile encoder calculates the source node profile $\mathbf{z}_i^{\mathrm{src}} \in \mathbb{R}^{d_{\mathrm{model}}}$ based on observations until $t_{\mathrm{current}}$, i.e., $f_{\mathrm{profile}}\left(s_i, \mathcal{H}_{t_{\mathrm{current}}}\right) = \mathbf{z}_i^{\mathrm{src}}$.

2. The message encoder encodes the interaction: $\mathbf{z}_i^{\mathrm{msg}} \in \mathbb{R}^{d_{\mathrm{model}}}$, i.e., $f_{\mathrm{msg}}\left(\mathbf{x}_i\right) = \mathbf{z}_i^{\mathrm{msg}}$.

3. The destination encoder generates destination embeddings: $\mathbf{z}_i^{\mathrm{dst}} \in \mathbb{R}^{d_{\mathrm{model}}}$, i.e., $f_{\mathrm{dst}}\left(d_i\right) = \mathbf{z}_i^{\mathrm{dst}}$.

4. The time encoder converts the interaction timestamp into a time embedding vector $\mathbf{z}_i^{\mathrm{t}} \in \mathbb{R}^{d_{\mathrm{model}}}$, i.e. $f_{\mathrm{time}}\left(t_i\right) = \mathbf{z}_i^{\mathrm{t}}$.

5. The readout layer predicts the interaction class $\hat{\mathbf{y}}_i$, i.e., $f_{\mathrm{rlc}}\left(\mathbf{z}_i^{\mathrm{src}}, \mathbf{z}_i^{\mathrm{msg}}, \mathbf{z}_i^{\mathrm{dst}}, \mathbf{z}_i^{\mathrm{t}},\right) = \hat{\mathbf{y}}_i$.

Once the predictions are made on mini-batch $\mathcal{E}_{\mathrm{batch}}$, the containers are updated so that they include the interactions in the mini-batch. The meta-paths are recalculated according to Equation 2. We note that in the procedure outlined above, it is only the construction of the source node profile that uses information from the historical interactions in the calculations. The destination node embedding is computed using only the identity of the node, and the message encoding is derived via a network operating on the message features. The destination embeddings are learnable, as are the weights of the message encoder network, so the historical interactions influence these embeddings via the training procedure. The overall flow is illustrated in Figure 2. Next we explain how individual modules are trained and describe the procedure used to update containers.

**Profile Encoder**  Inspired by our previous experiences working on webscale recommendation systems, we derive graphs that allow us to capture source-source correlations that might be obscured through traditional message passing schemes due to an over-smoothing effect. Similar to previous work, we define a meta-path as an edge that is constructed from a path through the graph (Chen & Lei, 2022; Huai et al., 2023; Huang et al., 2022). In our specific instance, we consider second-order meta-paths that connect a vertex which acts as a source to another which acts as a source through a shared destination vertex. The set of meta-paths is time dependent because the edges are parameterized by time. Given the set of meta-path edges $\mathcal{M}_{t_{\mathrm{current}}}$ observed up until the current time the profile encoder first builds the ego graph $\mathcal{G}_{t_{\mathrm{current}}}(s_i)$ over the set of vertices $\mathcal{V}_{t_{\mathrm{current}}}(s_i) = s_i \cup \{u : (u, s_i) \in \mathcal{M}_{t_{\mathrm{current}}}\}$ with set of edges $\mathcal{M}_{t_{\mathrm{current}}}(s_i) = \{(u, v) : (u, v) \in \mathcal{M}_{t_{\mathrm{current}}}; \forall u, v \in \mathcal{V}_{t_{\mathrm{current}}}(s_i)\}$. Thus, the ego graph $\mathcal{G}_{t_{\mathrm{current}}}(s_i) = (\mathcal{V}_{t_{\mathrm{current}}}(s_i), \mathcal{M}_{t_{\mathrm{current}}}(s_i))$ has a vertex set that consists of $s_i$ and all vertices that are con-

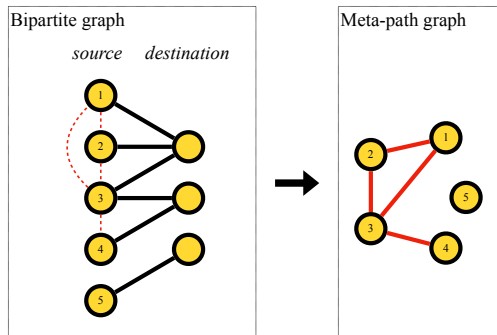

Figure 3: Meta-path construction: On the left, a bipartite graph is provided that consists of edges from a set of source entities to a set of destination entities. On the right, the final meta-path graph composed of meta-path edges is shown.

nected via a meta-path to $s_i$, and an edge set consisting of all meta-paths connecting these vertices. For each node $u \in \mathcal{V}_{t_{\mathrm{current}}}(s_i)$, we define the set of relevant event indices as $\mathcal{H}_{t_{\mathrm{current}}}(u) = \{j : s_j = u, \forall j \in \mathcal{H}_{t_{\mathrm{current}}}\}$.

The node embeddings are initialized by aggregating the embeddings of previous events associated with the corresponding node, i.e., $\mathbf{h}^{(0)}(u) = f_{\mathrm{aggregate}}(\mathcal{H}_{t_{\mathrm{current}}}(u))$. For example, using a mean aggregation schema with single layer of linear transformation, the node embeddings $\mathbf{h}^{(0)}(u) \in \mathbb{R}^{d_{\mathrm{model}}}$ are initialized as follows:

$$\mathbf{h}^{(0)}(u) = \frac{\sum_{i \in \mathcal{H}_{t_{\mathrm{current}}}(u)} \left[\mathbf{z}_i^{\mathrm{msg}} || \mathbf{z}_i^{\mathrm{dst}} || \mathbf{z}_i^{\mathrm{t}} || \mathbf{y}_i\right] \mathbf{W}_{\mathrm{event}}^{\mathrm{T}}}{|\mathcal{H}_{t_{\mathrm{current}}}(u)|}, \tag{3}$$

where $[\cdot || \cdot]$ denotes the concatenation operator, and $\mathbf{W}_{\mathrm{event}} \in \mathbb{R}^{d_{\mathrm{model}} \times d_1}$ are learnable weights for $d_1 = 3 \times d_{\mathrm{model}} + m$. Recall that $\mathbf{z}_i^{\mathrm{msg}}, \mathbf{z}_i^{\mathrm{dst}}, \mathbf{z}_i^{\mathrm{t}} \in \mathbb{R}^{d_{\mathrm{model}}}$ and $\mathbf{y}_i \in \mathbb{R}^m$. Then, using the GCN (Kipf & Welling, 2016) framework, at the $k^{\mathrm{th}}$ layer the node embeddings are updated by passing messages between neighbour nodes, i.e., $\mathbf{h}^{(k)}(u) = f_{\mathrm{gcn}}(\mathbf{h}^{(k-1)}(u), \mathcal{M}_{t_{\mathrm{current}}}(s_i))$. Introducing normalization coefficients $c_{u,v} = \frac{1}{\sqrt{\deg(u)} \cdot \sqrt{\deg(v)}}$, with $\deg(\cdot)$ denoting node degree on $\mathcal{G}_{t_{\mathrm{current}}}$, we can write the node embedding update at layer $k$ for a

normalized sum aggregation schema as:

$$\mathbf{h}^{(k)}(u) = \sum_{(u,v)\in\mathcal{M}_{t_{\text{current}}}(s_i)} c_{u,v}\left(\mathbf{h}^{(k-1)}(v)\mathbf{W}_k^{\mathrm{T}}\right), \tag{4}$$

where $\mathbf{W}_k \in \mathbb{R}^{d_{\text{model}}\times d_{\text{model}}}$ are learnable weights, and $\mathbf{h}^{(k)}(u) \in \mathbb{R}^{d_{\text{model}}}$. The profile embedding of source node $s_i$ is set to the final layer node embedding, i.e., $z_i^{\text{src}} = \mathbf{h}^{(K)}(s_i)$ where $K$ denotes the total number of message passing layers.

**Time Encoder**  For the time encoder, we employ either the fixed time encoding function proposed by Cong et al. (2023) or the learnable time projection introduced by Kumar et al. (2019). Given a weight vector $\omega \in \mathbb{R}^{d_{\text{model}}}$, in general the time encoding function follows:

$$f_{\text{time}}(t) = \cos(\omega t) = \mathbf{z}^t, \tag{5}$$

where $\mathbf{z}^t \in \mathbb{R}^{d_{\text{model}}}$ denotes the vector representation of timestamp $t$. The two variants of time encoding we investigate in this work differ in the calculation of the weight vector. In the *learnable version*, $\omega \in \mathbb{R}^{d_{\text{model}}}$ is simply learned during training, so the time encoding layer is a linear projection, without any bias, followed by cosine scaling. In the case of *fixed time encoding* as proposed by Cong et al. (2023), each dimension of the weight vector is set to $\omega_i = \alpha - \frac{(i-1)}{\beta}$, where $\alpha$ and $\beta$ are scalars, so that $\omega t$ is a vector with monotonically exponentially decreasing values. The hyperparameters $\alpha$ and $\beta$ are selected according to the scale of the minimum and maximum timestamps in the data. In practice $\alpha = \beta = \sqrt{d_{\text{model}}}$ is found to perform well, and we follow this setting in our experiments.

**Other Modules**  The message encoder uses a single linear layer to compute $f_{\text{msg}}(\mathbf{x}_i) = \mathbf{x}_i\mathbf{W}_{\text{msg}}^{\mathrm{T}} + \mathbf{b}_{\text{msg}}$, where $\mathbf{W}_{\text{msg}} \in \mathbb{R}^{d_{\text{model}}\times d_{\text{msg}}}$ are learnable weights and $\mathbf{b}_{\text{msg}} \in \mathbb{R}^{d_{\text{model}}}$ is a learnable bias. For destination encoding, we use an embedding look-up table of size equal to the number of destination nodes. This is initialized randomly, $f_{\text{dst}}(d_i) = \mathbb{1}_{d_i}\mathbf{W}_{\text{dst}}^{\mathrm{T}}$, where $\mathbb{1}_{d_i} \in \mathbb{R}^{||\mathcal{V}_{\text{dst}}||}$ denotes a one-hot vector representation of node $d$, and $\mathbf{W}_{\text{dst}} \in \mathbb{R}^{d_{\text{model}}\times ||\mathcal{V}_{\text{dst}}||}$ are learnable weights. The predictions at the readout layer are computed as $\hat{\mathbf{y}}_i = \left[\mathbf{z}_i^{\text{src}} + \mathbf{z}_i^{\text{dst}} + \mathbf{z}_i^{\text{msg}} + \mathbf{z}_i^t\right]\mathbf{W}_{\text{rlc}}^{\mathrm{T}}$ where $\mathbf{W}_{\text{rlc}} \in \mathbb{R}^{d_{\text{model}}\times d_{\text{model}}}$ are learnable weights.

Unlike static graph learning methods, Graph Profiler is capable of encoding temporal properties of network. Graph Profiler has two main advantages over existing TGL methods: (1) it enables the construction of dynamic entity profiles that take into account feature and label information of previous interaction in the neighbourhood; and (2) it is capable of maintaining a long-term view of an entity's profile that can capture long-term preferences. In addition, the modular structure of Graph Profiler is flexible, so the model can easily be adapted to suit the contextual properties of individual datasets.

## 5  Experiments

In order to understand RLC as a novel task within TGL, we begin by evaluating a two layer Multi-layer Perceptron (MLP), TGN (Rossi et al., 2020), TGAT (da Xu et al., 2020), CAWN (Wang et al., 2021) and Graph Mixer (Cong et al., 2023) on RLC by making the appropriate modifications to the algorithms. We have chosen these methods because they are state-of-the-art TGL baselines developed for the FLP task. Based on our observations concerning the performance of the methods, we outline a set of design principles that comprise the design space for RLC. With these design principles in mind, we present Graph Profiler and benchmark it on six different datasets. For each dataset, we locate the optimal portion of our design space and discuss the correspondence between that and the underlying dynamics of the dataset under investigation.

**Datasets**  We evaluated our methods on four benchmark datasets that have previously been used by the TGL community – YELPCHI (Dou et al., 2020), WIKIPEDIA, MOOC, and REDDIT (Kumar et al., 2019). These datasets are usually employed as benchmarks for future link prediction; we adopt suitable measures to convert them to the RLC setting. In YELPCHI, the source entities are platform users and the destination entities include hotels and restaurants. An interaction happens when a user reviews one of the hotels or restaurants. The reviews are labeled either as filtered (spam) or recommended (legitimate). For WIKIPEDIA,

Table 1: Dataset statistics. $|\mathcal{V}_{\text{src}}|$ and $|\mathcal{V}_{\text{dst}}|$ denote the number of source entities and destination entities, respectively; $d_{\text{msg}}$ denotes the number of interaction features; $|\mathcal{E}|$ denotes the number of interactions; $\rho$ denotes the fraction of edge labels that are positive; $\bar{\mathcal{H}}_e$ denotes the average edge homophily; $\bar{\mathcal{H}}_e^+$ and $\bar{\mathcal{H}}_e^-$ denote the edge homophily for positive and negative classes, respectively, and $\tilde{\mathcal{H}}_e^b$ denotes the balanced edge homophily. Edge homophily metrics are calculated over all interactions (edges) in the dataset.

| | $|\mathcal{V}_{\text{src}}|$ | $|\mathcal{V}_{\text{dst}}|$ | $|\mathcal{E}|$ | $d_{\text{msg}}$ | $\rho$ | $\bar{\mathcal{H}}_e$ | $\mathcal{H}_e^+$ | $\mathcal{H}_e^-$ | $\bar{\mathcal{H}}_e$ |
|---|---|---|---|---|---|---|---|---|---|
| EPIC GAMES | 542 | 614 | 17584 | 400 | 0.6601 | 0.8330 | 0.9038 | 0.6955 | 0.7663 |
| YELPCHI | 38,063 | 201 | 67,395 | 25 | 0.1323 | 0.7781 | 0.1589 | 0.8725 | 0.2533 |
| WIKIPEDIA | 8,227 | 1,000 | 157,474 | 172 | 0.0014 | 0.9975 | 0.0130 | 0.9988 | 0.0144 |
| MOOC | 7,047 | 97 | 411,749 | 4 | 0.0099 | 0.9809 | 0.0212 | 0.9904 | 0.0308 |
| REDDIT | 10,000 | 984 | 672,447 | 172 | 0.0005 | 0.9989 | 0.0025 | 0.9995 | 0.0030 |
| OPEN SEA | 57,230 | 1,000 | 282,743 | 35 | 0.4601 | 0.5865 | 0.5505 | 0.6171 | 0.5812 |

the set of entities is composed of users and pages, and an interaction happens when a user edits a page. For REDDIT, entities are users and subreddits, and an interaction represents a post written by a user on a subreddit. Some page edits on Wikipedia and posts on Reddit may be controversial causing the user to be banned. Thus, on both datasets we base the target class of interaction on whether it is controversial or not, i.e., whether the user was banned as a result of posting it. The interaction features for these three datasets are extracted based on the textual content of edit/post/review. The MOOC dataset consists of actions performed by students on a MOOC online course. The source entities are the students and the destination entities are the course contents that the students interact with, e.g., recordings or lecture notes. The interaction features are the types of activities the student performed during an interaction, e.g., viewing the video or submitting an answer on the forum. Sometimes, the students drop out of the course after an activity. We use this as a label to identify the target class of an interaction. Thus, all four datasets are binary recent link classification datasets for which the class imbalance is salient.

In addition to adapting benchmark TGL datasets to RLC task, we process two tabular datasets that are not conventionally investigated in TGL setting; EPIC GAMES [1] and OPEN SEA (La Cava et al., 2023a;b; Costa et al., 2023) [2]. The Epic Games Store is a digital video game storefront, operated by Epic Games. The dataset includes the critiques written by different authors about the games released on the platform. The source and destination nodes represent authors and games, respectively, and the critiques form the set of interactions. We construct the interaction features by vectorizing the textual content of critiques using TF-IDF. We include the overall rating the author provided as an additional interaction feature. The label of the interaction is determined based on whether it was selected as top critique or not. Once a critique is released all the information regarding the author, game and features of the critique is available. Its selection as a top-critique is a delayed observation, so the data naturally forms an RLC task. Open Sea is one of the leading trading platforms in the Web3 ecosystem. The dataset is a collection of Non-Fungible Token (NFT) transactions. Sourced from Open Sea, it is provided as a natural language processing dataset and is mainly used for multimodal learning classification tasks. To the best of our knowledge the dataset has not been previously investigated in a TGL framework. In order to make it amenable to graph learning, we identify disjoint sets of sellers and buyers of unique NFTs (these are identified by collection memberships and token IDs) to serve as source and destination nodes. The transaction features are a binary representation of categorical variable fields associated with the transaction, the cryptocurrency exchange rates at the time of the interaction and the associated monetary values. The label of a transaction is determined based on a future transaction of the unique NFT. It is tagged as 'profitable' if the revenue obtained through the final sale is higher than the price paid at the purchase. The labels are thus delayed, because whether it will be a profitable investment is not known at the time of purchase. The data thus aligns with the RLC setting. In Appendix B, further details on pre-processing of OPEN SEA dataset is shared. The datasets statistics are provided in Table 1. In our experiments, data is divided into training (70%), validation (10%) and testing (20%) sets chronologically.

---

[1] https://www.kaggle.com/datasets/mexwell/epic-games-store-dataset.
[2] https://huggingface.co/datasets/MLNTeam-Unical/NFT-70M_transactions.

**Edge Homophily** We introduce a measure of edge homophily to understand the importance of graph information to our edge classification task. Denote by $\mathcal{N}^e(\alpha)$ an edge-wise neighbourhood operator that constructs a set of all edges that are connected to a given edge, $\alpha = (i, j)$, where $i$ and is the source and $j$ the destination. This operator forms the union of two sets, i.e., $\mathcal{N}^e(\alpha) = \mathcal{I}(i) \cup \mathcal{O}(j)$, where $\mathcal{I}(i)$ is the set of incoming edges connected to the source $i$ and $\mathcal{O}(j)$ is the set of outgoing edges connected to the destination. Our edge homophily measure is then defined as:

$$\bar{\mathcal{H}}_e(\mathcal{G}) = \frac{1}{|\mathcal{E}|} \sum_{\alpha \in \mathcal{E}} \sum_{\beta \in \mathcal{N}_\alpha^{(e)}} \frac{\mathbf{1}_{l(\alpha)=l(\beta)}}{|\mathcal{N}_\alpha^{(e)}|}, \tag{6}$$

where $\mathcal{N}^e$ is the edge-wise neighbourhood operator and $l$ is the operator that returns the label of the edge. The edge-wise neighbourhood operator constructs a set of all edges that are connected to a given edge, $\alpha = (i, j)$, where $i$ and $j$ are the source and destination respectively, by constructing the union of two sets $\mathcal{N}^e(\alpha) = \mathcal{I}(i) \cup \mathcal{O}(j)$, where $\mathcal{I}(\cdot)$ and $\mathcal{O}(\cdot)$ construct the set of incoming and outgoing edges respectively. For the simplicity of notation, we have neglected the time dimension but this definition is easy to generalize to temporal graphs through the neighbourhood operators. Edge-homophily measures the fraction of neighbouring edges that have the same class, in analogy to the way node-homophily measures the fraction of neighbouring nodes with the same class. Node-homophily is an important dataset property that can be highly indicative of the value that can be derived by encoding graph structure in node classification tasks (Pei et al., 2020), particularly for embedding procedures that rely on smoothing over a neighbourhood. The edge homophily definition in Equation 6 treats different classes equally, which can be misleading for imbalanced datasets. Therefore, using $\rho$ to denote the fraction of positive edge labels, we also define a balanced edge homophily metric for binary classification as $\tilde{\mathcal{H}}_e^b(\mathcal{G}) = (1 - \rho)\mathcal{H}_e^+(\mathcal{G}) + \rho\mathcal{H}_e^-(\mathcal{G})$, where $\bar{\mathcal{H}}_e^+(\mathcal{G}) = \frac{1}{|\mathcal{E}^+|} \sum_{\alpha \in \mathcal{E}^+} \sum_{\beta \in \mathcal{N}_\alpha^{(e)}} \frac{\mathbf{1}_{l(\alpha)=l(\beta)}}{|\mathcal{N}_\alpha^{(e)}|}$ and $\bar{\mathcal{H}}_e^-(\mathcal{G}) = \frac{1}{|\mathcal{E}^-|} \sum_{\alpha \in \mathcal{E}^-} \sum_{\beta \in \mathcal{N}_\alpha^{(e)}} \frac{\mathbf{1}_{l(\alpha)=l(\beta)}}{|\mathcal{N}_\alpha^{(e)}|}$. Table 1 presents the edge homophily values for the datasets we study. In Appendix C, we illustrate the dynamics of edge homophily over time. In the case of WIKIPEDIA, MOOC, and REDDIT, the overall edge homophily reaches as high as 99%, whereas the homophily among positive class edges falls below 5%, a phenomenon attributed to significant class imbalance. Across these three platforms, there is a noticeable trend of decreasing positive edge homophily over time, a pattern that emerges as banned users or students who withdraw from courses exit the network. Conversely, on YELPCHI, there's an observable increase in positive class homophily and a decrease in negative class homophily. This trend could stem from spam attacks on businesses, typically carried out by similar groups of reviewers whose numbers grow over time. A similar pattern is detected on Open Sea, suggesting an increasing ratio of sellers who profit over time. Nonetheless, those who profited in the system's early stages remain active participants.

**Performance Evaluation** Two of the most common metrics used for performance evaluation for both FLP and DNC are area under the receiver operating characteristic curve (AUC) and average precision score (APS). These metrics exhibit some undesirable behaviour. AUC is known to saturate, such that it become impossible to differentiate between candidate algorithms. Both metrics can provide skewed measures of quality when applied to imbalanced datasets (Chicco & Jurman, 2020; 2023). As a result, we turn to the *Matthews Correlation Coefficient (MCC)* (Yule, 1912; Gorodkin, 2004), which is defined as:

$$\text{MCC} = \frac{cs - \vec{t} \cdot \vec{p}}{\sqrt{s^2 - \vec{p} \cdot \vec{p}}\sqrt{s^2 - \vec{t} \cdot \vec{t}}}, \tag{7}$$

where $\vec{t}$ is a vector, with each element being the number of times a class occurred, $\vec{p}$ is a vector of the number of times each class is predicted, $c$ is the number of samples correctly predicted, and $s$ is the total number of samples. This correlation coefficient has a maximum value of 1, and the minimum value ranges between -1 and 0, depending on the distribution of the underlying data. A score of 0 indicates that the predictor is perfectly random; a score of 1 indicates that the predictor is perfectly accurate; and a score of -1 indicates that the predictor is perfectly inaccurate. As an illustrative example, we present use case A1 from Table 4 in Chicco & Jurman (2020). In this example, we have 100 total data points with 91 in the positive class and 9 in the negative. For a hypothetical classifier that that predicts all but one data point as a member of the positive class; we find $TP = 90, FN = 1, TN = 0, FP = 9$. This yields a respectable APS of 0.90 but a near

Table 2: Performance of TGN variants. The performance of TGN variants is presented as the average of 10 random seed runs for each configuration. The best time encoding and aggregator versions are highlighted in bold and the cells representing readout configurations are shaded based on their values, with darker shading indicating higher performance.

| | | YelpCHI | | | Wikipedia | | | Mooc | | | Reddit | | |
|---|---|---|---|---|---|---|---|---|---|---|---|---|---|
| | | MCC | APS | AUC | MCC | APS | AUC | MCC | APS | AUC | MCC | APS | AUC |
| Time Encoding | fix | 0.2624 | 0.3148 | 0.7590 | **0.2943** | **0.1237** | **0.9086** | **0.1004** | 0.0486 | 0.7634 | 0.0042 | 0.0049 | **0.6608** |
| | learn | **0.2866** | **0.3278** | **0.7723** | 0.1933 | 0.0989 | 0.8728 | 0.0973 | **0.0571** | **0.7730** | **0.0444** | **0.0093** | 0.6508 |
| Aggregator | exp | 0.2803 | 0.3262 | 0.7700 | 0.1018 | 0.0712 | 0.8653 | **0.0630** | **0.0415** | **0.7494** | 0.0158 | 0.0036 | **0.6608** |
| | last | **0.2866** | **0.3278** | **0.7723** | **0.2943** | **0.1237** | **0.9086** | 0.0477 | 0.0325 | 0.7045 | **0.0444** | 0.0055 | **0.6599** |
| | mean | 0.2744 | 0.3217 | 0.7666 | 0.2034 | 0.0896 | 0.8955 | 0.1004 | 0.0571 | 0.7730 | 0.0142 | **0.0093** | 0.6235 |
| Readout | src | 0.2286 | 0.2391 | 0.7096 | 0.1237 | 0.0828 | 0.8368 | 0.0530 | 0.0416 | 0.7199 | 0.0105 | 0.0045 | 0.6435 |
| | dst | 0.2288 | 0.2311 | 0.7015 | 0.0972 | 0.0464 | 0.7298 | 0.0432 | 0.0377 | 0.7195 | 0.0099 | 0.0049 | 0.6188 |
| | src-dst | 0.2319 | 0.2411 | 0.7094 | 0.1018 | 0.0355 | 0.8908 | 0.0924 | 0.0462 | 0.7485 | 0.0142 | 0.0031 | 0.6608 |
| | src-t | 0.2311 | 0.2426 | 0.7147 | 0.1308 | 0.0844 | 0.8401 | 0.0507 | 0.0386 | 0.7191 | 0.0104 | 0.0021 | 0.6472 |
| | dst-t | 0.2277 | 0.2381 | 0.7063 | 0.1057 | 0.0469 | 0.7379 | 0.0338 | 0.0411 | 0.7205 | 0.0159 | 0.0124 | 0.6211 |
| | src-dst-t | 0.2290 | 0.2542 | 0.7151 | 0.1442 | 0.0346 | 0.9086 | 0.0903 | 0.0506 | 0.7729 | 0.0444 | 0.0092 | 0.6508 |
| | src-msg | 0.2732 | 0.3166 | 0.7627 | 0.1530 | 0.0835 | 0.8808 | 0.0996 | 0.0603 | 0.7763 | 0.0074 | 0.0024 | 0.6046 |
| | dst-msg | 0.2744 | 0.3209 | 0.7641 | 0.1184 | 0.0564 | 0.8332 | 0.0475 | 0.0289 | 0.7112 | 0.0171 | 0.0144 | 0.5987 |
| | src-dst-msg | 0.2644 | 0.3147 | 0.7629 | 0.2040 | 0.0714 | 0.8537 | 0.0894 | 0.0497 | 0.7708 | 0.0158 | 0.0033 | 0.6599 |
| | src-msg-t | 0.2866 | 0.3278 | 0.7723 | 0.1764 | 0.0858 | 0.8994 | 0.1004 | 0.0571 | 0.7727 | 0.0040 | 0.0014 | 0.6005 |
| | dst-msg-t | 0.2803 | 0.3230 | 0.7669 | 0.1300 | 0.0617 | 0.7489 | 0.0462 | 0.0325 | 0.7045 | 0.0113 | 0.0093 | 0.5971 |
| | src-dst-msg-t | 0.2734 | 0.3217 | 0.7666 | 0.2943 | 0.1237 | 0.9020 | 0.0973 | 0.0536 | 0.7730 | 0.0077 | 0.0049 | 0.6089 |

random MCC of -0.03. While simple, this is just one example where metrics like APS can mask underlying poor performance for imbalanced datasets. Chicco & Jurman (2023) further presents similar failure modes for AUC. Because of this, we choose MCC as our figure of merit for the RLC tasks that we present.

### 5.1 Key factors to tailor model to specific needs of data

With the introduction of RLC as a benchmark task to evaluate TGL methods, we explore the performance of a well-known state-of-the art model, TGN (Rossi et al., 2020), together with variants. We create these variants by constructing different message aggregation schema, readout layers, and time encoding strategies to better discover the potential of the overall architecture. More specifically, we experiment with (1) fixed and learnable time encoding as proposed by (Cong et al., 2023) and by (Kumar et al., 2019), respectively; (2) mean, last, and exponential decay message aggregators; and (3) six different configurations of the input to the readout layer based on different combinations of source, destination, time and message embeddings calculated for the most recent event. As described by (Rossi et al., 2020), the mean aggregator calculates the state of a node by averaging the interactions held in the memory, whereas the last aggregator uses only the most recent interaction. The exponential decay variant calculates a weighted average over the interactions in the memory by setting weights such that they decrease exponentially with respect to the time that has elapsed. Table 2, summarizes the results. We present violin plots for each variant in Appendix E.

The readout variations appeared to play a significant role in the model's performance, as can be seen in Figure 4. These results on the Wikipedia dataset are demonstrated for different model dimensions, i.e. $d \in \{100, 200\}$ and a varying number of neighbours. We observe that incorporating the edge features as a residual at the final layer of the update helps to improve the performance in terms of MCC, which makes intuitive sense given that the message features for this dataset correspond to the edit's content. Interestingly, while we observe this trend when looking at the MCC curves, which exhibit a dramatic stratification in performance, the AUC curves show the opposite trend. We attribute this to AUC being an unsatisfactory metric for the evaluation of RLC tasks, particularly if there is class imbalance. The APS and Loss curves exhibit trends that are similar to those of MCC (See Appendix F). We conclude that for abuse-related datasets the most recent interaction matters most, and therefore aggregation based on the last event is more useful. In the case of predicting course completion, averaging over multiple previous actions is valuable, which is captured by the outperformance of the mean aggregator. In general, we observe that involving recent interaction features in the readout layer is very useful. The configurations with `msg` included perform significantly better.

Table 3: Model comparison results. The performance of models is presented as the average of 10 random seed runs. The best and second best results are highlighted in red and blue, respectively.

|     |               | EPIC GAMES | YELPCHI | WIKIPEDIA | MOOC   | REDDIT | OPEN SEA |
|-----|---------------|-----------|---------|-----------|--------|--------|----------|
| MCC | MLP           | 0.1554    | 0.2763  | 0.2354    | 0.0673 | 0.0021 | 0.1106   |
|     | TGN           | 0.8373    | 0.2372  | 0.1442    | 0.0991 | 0.0174 | 0.1071   |
|     | TGAT          | 0.5546    | 0.1890  | 0.0000    | 0.0000 | 0.0000 | 0.2245   |
|     | CAWN          | 0.4925    | 0.2300  | 0.0000    | 0.0103 | 0.0000 | 0.3223   |
|     | Graph Mixer   | 0.2316    | 0.2830  | 0.1442    | 0.1174 | 0.0000 | 0.2647   |
|     | Modified TGN  | 0.8713    | 0.2866  | 0.2943    | 0.1004 | 0.0444 | 0.2647   |
|     | Graph Profiler| 0.9355    | 0.3274  | 0.2498    | 0.1739 | 0.0115 | 0.2959   |
| APS | MLP           | 0.6976    | 0.3254  | 0.0759    | 0.0169 | 0.0012 | 0.7790   |
|     | TGN           | 0.9850    | 0.2461  | 0.0320    | 0.0526 | 0.0018 | 0.7912   |
|     | TGAT          | 0.8844    | 0.1789  | 0.0157    | 0.0190 | 0.0014 | 0.6582   |
|     | CAWN          | 0.8441    | 0.2168  | 0.0269    | 0.0367 | 0.0014 | 0.7247   |
|     | Graph Mixer   | 0.7522    | 0.3252  | 0.0550    | 0.0711 | 0.0021 | 0.8304   |
|     | Modified TGN  | 0.9892    | 0.3249  | 0.1148    | 0.0430 | 0.0055 | 0.8225   |
|     | Graph Profiler| 0.9988    | 0.4059  | 0.0955    | 0.0896 | 0.0092 | 0.8459   |
| AUC | MLP           | 0.6117    | 0.7694  | 0.7731    | 0.6447 | 0.5486 | 0.5776   |
|     | TGN           | 0.9734    | 0.7135  | 0.7135    | 0.7672 | 0.5671 | 0.5908   |
|     | TGAT          | 0.8470    | 0.6314  | 0.8908    | 0.6383 | 0.5336 | 0.6871   |
|     | CAWN          | 0.7931    | 0.6850  | 0.8880    | 0.7107 | 0.6100 | 0.6994   |
|     | Graph Mixer   | 0.7132    | 0.7650  | 0.7500    | 0.7515 | 0.6413 | 0.6789   |
|     | Modified TGN  | 0.9807    | 0.7723  | 0.7723    | 0.7439 | 0.6508 | 0.6544   |
|     | Graph Profiler| 0.9974    | 0.8058  | 0.7821    | 0.7886 | 0.6280 | 0.6740   |

## 5.2 Model Comparison

Using the results from the TGN modifications, we have identified multiple core design principles associated with improved performance on the RLC task, and we have incorporated these principles into Graph Profiler. Specifically, we observed that a learnable time-encoding provided improved results for three of four data sets. The `src-dst-msg-t` readout variant provided strong results across all four data sets. Because of these results, we designed Graph Profiler with a learnable time encoder and a `src-dst-msg-t` readout function. To validate these results, we perform benchmark experiments of baselines and Graph Profiler on RLC. For

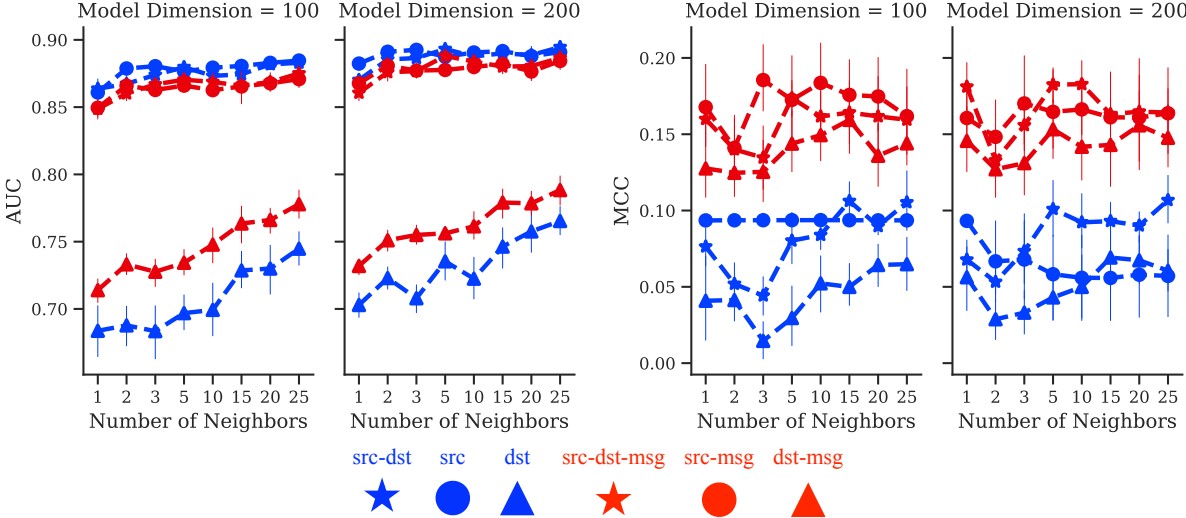

Figure 4: Readout variations on Wikipedia. The blue glyphs correspond to combinations of the vertex features, while the red glyphs correspond to combinations of the vertex *and* message features. The star, circle, and triangle glyphs correspond to the src-dst, src, and dst embeddings respectively.

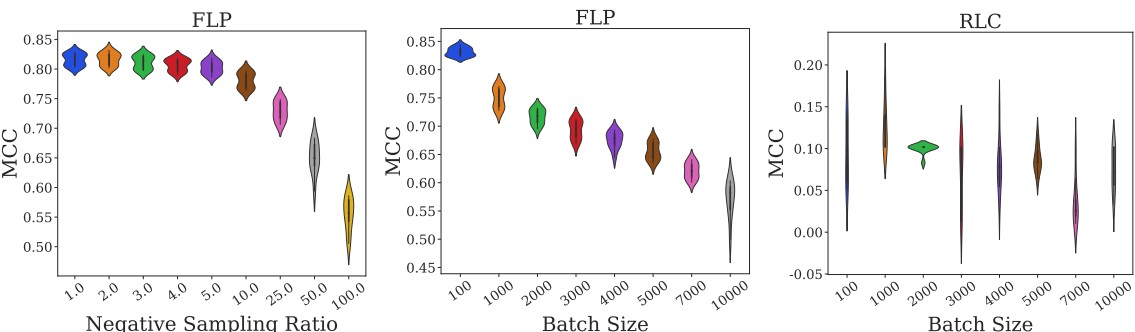

Figure 5: TGN performance on FLP vs RLC on Wikipedia with varying levels of negative sampling ratio and batch size.

the existing benchmark datasets, Graph Profiler obtains the best results for YELPCHI and MOOC, but our modified version of the TGN architecture outperforms for WIKIPEDIA and REDDIT. The most probable reason is that the ratio of the number of source nodes to the number of destination nodes is much higher for YELPCHI and MOOC compared to WIKIPEDIA and REDDIT. Graph Profiler constructs embeddings for the source nodes via calculations over neighbouring interactions, but uses learnable embeddings in a lookup table for the destination nodes. In the majority of bipartite graphs examined, there is a significantly larger number of source nodes compared to destination nodes. This discrepancy allows for the practicality of learning a unique embedding vector for each destination node.

Another observation we draw from these model comparison results is the usefulness of MCC in highlighting the relative capabilities of different models. For example, on MOOC Graph Profiler improves MCC by 73% compared to the Modified TGN, while the observed change for the AUC is only 3% (See Table 3). On the EPIC GAMES, which is less imbalanced, Graph Profiler outperforms other baselines based on MCC and APS, but the performance of the techniques cannot be reliably distinguished using AUC. On OPEN SEA the Graph Profiler and CAWN per par in terms of MCC and APS. Revisiting the dataset statistics provided in Table 1, we conclude that encoding node profiles based on their shared history, i.e., employing the Graph Profiler architecture, is more effective on datasets with higher balanced edge homophily and less class imbalance. For such datasets, tracking a graph-wide memory, i.e., employing the TGN framework, is less effective.

### 5.3 On the importance of hyperparameter sensitivity differences between FLP and RLC

In our reproduction study we explored the effect of negative sampling ratio, batch size, and the number of sampled neighbors on the performance of our TGN baseline for the FLP task. We term these as non-architectural parameters because they influence the training but do not influence the architecture of the model itself. We explore these parameters because they impact the trade-off between model accuracy versus computational performance and utilization. In the example of batch size, this is typically tuned to be as large as possible to maximize GPU utilization, but we see in Figure 5, a steady decline in MCC as the batch size is increased (See Appendix D for other metrics). Indeed, we observe variation in model performance due to changes in batch size that are larger than the variations across different model architectures. Intuitively, the decay makes sense, because the gradient updates become less frequent, but this points to a relatively significant but under-discussed trade-off that has major ramifications for production use-cases. In the case of the negative sampling ratio, we observe a slight decline in MCC and a decrease in the consistency between individual training runs as the number of negative samples increases. Thus, it can be concluded that RLC does not have the same dependence on batch-size and does not require the generation of negative samples. These observations lead us to the conclusion that (1) the assumptions made during the development of models for FLP may not hold for RLC, and direct translation of existing TGL methods, which are generally benchmarked on FLP tasks, to perform RLC in industrial settings is not advisable, and (2) RLC is an interesting general purpose benchmark task for the TGL community, and should be treated differently from the current common tasks.

### 5.4 Ablation on Learnable Destination Embeddings

In order to investigate the impact of learnable destination embeddings, we experiment with and without learning a set of embeddings for the destination nodes. The results are presented in Table 4. For most of the datasets (with the exception of RED-DIT), incorporating learnable destination node representations improves the predictive performance of Graph Profiler. The destination node embeddings

Table 4: Impact of learnable destination embeddings

|  | MCC | | APS | | AUC | |
|---|---|---|---|---|---|---|
|  | *with* | *without* | *with* | *without* | *with* | *without* |
| Epic Games | **0.9355** | 0.7695 | **0.9988** | 0.9575 | **0.9974** | 0.9054 |
| YelpCHI | **0.3274** | 0.3071 | **0.4059** | 0.3892 | **0.8058** | 0.8000 |
| Wikipedia | **0.2498** | 0.1324 | **0.0955** | 0.0366 | **0.7821** | 0.6946 |
| Mooc | **0.1739** | 0.0000 | **0.0896** | 0.0011 | **0.7886** | 0.5906 |
| Reddit | 0.0115 | **0.0701** | 0.0092 | **0.0203** | 0.6280 | **0.6833** |

are then used for building the source node profiles. It can be inferred that destination encoding enriches the source profile embeddings by temporally smoothing the interactions to build a sense of history. By contrast, using the destination in the readout itself means that the history of destination nodes is ignored. We believe that the difference for the REDDIT dataset may arise from the fact that the labels (whether a sub-reddit is controversial) is less dependent on the main post, compared to other Wikipedia. Said another way, there are many Wikipedia pages that are prone to abuse for political reasons, and thus, the page profile (destination node) matters. By contrast, abuse on Reddit is less dependent on the sub-reddit (destination node) than the author (source node).

### 5.5 Ablation Study on Graph Profiler Components

To assess the significance of different components of Graph Profiler, we conducted an ablation study. This involved systematically removing specific module from the model and comparing the outcomes with those of the intact version. For a more in-depth comparative analysis, we utilized datasets from YELPCHI and EPIC GAMES. These were chosen for their comprehensive content and manageable sizes, facilitating the intensive computations required for a thorough statistical evaluation. The findings are presented in Table 5, indicating that while each component plays a critical role, their significance varies across contexts. For instance, the profile encoder proves to be exceptionally valuable for the EPIC GAMES dataset, whereas encoding features of recent interactions has a more pronounced effect on the YELPCHI dataset. This suggests that, in the context of the EPIC GAMES platform, the profile information of reviewers holds greater predictive power for identifying top critiques than the content of their reviews. Conversely, for YELPCHI, the content of reviews is more crucial in detecting spam.

Table 5: Impact of individual components. Each row reports the results with the configuration **without** the indicated module. Results are averaged over 10 different random seeds.

|  | Epic Games | YelpCHI | Epic Games | YelpCHI | Epic Games | YelpCHI |
|---|---|---|---|---|---|---|
|  | MCC | | APS | | AUC | |
| Graph Profiler | 0.9355 | 0.3274 | 0.9988 | 0.4059 | 0.9974 | 0.8058 |
| w.o. **Profile Encoder** | 0.0000 | 0.2849 | 0.6097 | 0.3316 | 0.5000 | 0.7725 |
| w.o. **Learning Destination Embeddings** | 0.7695 | 0.3071 | 0.9575 | 0.3892 | 0.9054 | 0.8000 |
| w.o. **Recent Event Message Encoder** | 0.6311 | 0.1926 | 0.8940 | 0.2215 | 0.8593 | 0.6568 |
| w.o. **Recent Event Time Encoder** | 0.8082 | 0.3072 | 0.9766 | 0.3970 | 0.9590 | 0.7986 |

## 6 Conclusion

In this work, we introduce *Recent Link Classification (RLC)* on temporal graphs as a benchmark downstream task and evaluate the most competitive state-of-the-art method's performance using a statistically meaningful metric, namely *Matthews Correlation Coefficient (MCC)*, which is more robust to imbalanced datasets in comparison to the commonly used metrics. We propose several design principles, which involve the choice of message aggregation schema, readout layer and time encoding strategy, for tailoring models to specific requirements of the task and the dataset. We show that appropriate selection can lead to a significant improvement on benchmark datasets. We present, Graph Profiler, an RLC algorithm designed for bipartite graphs which are commonly encountered in industrial settings. We believe the introduction of RLC as a

benchmark task for temporal graph learning is useful for the evaluation of prospective methods within the field.

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

## Appendices

### A    Implementation Details

In an effort to present fair comparisons, we performed 100 steps of hyperparameters optimization to optimize the hyperparameters of all models using the software package OPTUNA (Akiba et al., 2019). Each experiment was run over the same 10 seeds. All tuning was performed on the validation set where we maximize the average accuracy across all 10 seeds, and we report the test-seed averaged results on the test set that are associated with those hyperparameter settings that maximize the validation accuracy. All models were implemented using PYTORCH GEOMETRIC 2.3.1 (Fey & Lenssen, 2019) and PYTORCH 1.13 (Paszke et al., 2019). We implemented TGN using the layers that are publicly available in PYTORCH GEOMETRIC, GraphMixer [3] and TGAT [4] were implemented using the authors opensource implementation provided on their github repository. All computations were run on an `Nvidia DGX A100` machine with 128 `AMD Rome 7742` cores and 8 `Nvidia A100` GPUs.

### B    Details of Pre-processing on Open Sea Dataset

The OPEN SEA dataset is a collection of Non-Fungible Token (NFT) transactions conducted between 2021 and 2023, provided by La Cava et al. (2023a); Costa et al. (2023); La Cava et al. (2023b). Originally, the data involves 70 million transactions chronologically divided into ten splits. In order to scale our experiments, we use the first split which is composed of 7,097,215 transactions spanned between 2022-12-17 and 2023-02-18. In Table 6 the fields of the raw data is provided. The pre-processing steps are as follows:

1. The unique NFT ids are set grouping the transaction by `token_id` and `collection_name`.

2. The transactions are filtered for each unique NFT, and assigned an `order` based on their position in the sequence and the last ones are flagged. A new field of `next_revenue` is created such that for $k^{\text{th}}$ transaction in the sequence is assigned with the `usd_gain` of $(k+1)^{\text{th}}$ one, except the flagged ones.

3. For each transaction without the flag, a new field of `benefit` is calculated by `next_revenue - usd_price` and the flagged transactions are filtered.

4. The number of transactions reduces to 2,979,950 which is still high for the scale of dataset size we experiment with. Thus, we select 1000 different buyers by `winner_account` who conduced most transactions and filter transactions that involve them, which reduces the number of transactions to 282,743.

5. The source node identities are set by `seller_account`, destination node identities by `winner_account`, the transaction time is set by `tx_timestamp`.

6. Feature vectors of transactions are calculated by concatenating normalized quantitative feature fields [`fees_seller`, `fees_opensea`, `fees_seller_usd`, `fees_opensea_usd`, `price`, `gain`, `usd_price`, `usd_gain`, `to_eth`, `to_usd`, `created_date`] and binary representations of categorical feature fields [`token`, `chain`, `token_type`, `asset_contract_type`, `asset_type`].

7. The response variable is set by binarizing `benefit` such that it is set to 1 if `benefit` $> 0$, and 0 otherwise.

---

[3] https://github.com/CongWeilin/GraphMixer
[4] https://github.com/StatsDLMathsRecomSys/Inductive-representation-learning-on-temporal-graphs

Table 6: Raw data fields of OPEN SEA dataset

| Variable | Type | Description | Processing |
|---|---|---|---|
| token_id | String | The id of the NFT — this value is unique within the same collection | Anonymized |
| num_sales | Integer | A progressive integer indicating the number of successful transactions involving the NFT up to the current timestamp (cf. *tx_timestamp*) | Original |
| nft_name | Vector ID | The name of the NFT | Anonymized |
| nft_description | Vector ID | The description of the NFT as provided by the creator | Anonymized |
| nft_image | Vector ID | The ID for accessing the NFT image vector | Anonymized |
| collection_name | Vector ID | The ID for accessing the Collection name vector | Anonymized |
| collection_description | Vector ID | The ID for accessing the Collection description vector | Anonymized |
| collection_image | Vector ID | The ID for accessing the Collection image vector | Anonymized |
| fees_seller | Float | The absolute amount of fees the seller has gained from this transaction expressed in *token* | Original |
| fees_opensea | Float | The absolute amount of fees OpenSea has gained from this transaction expressed in *token* | Original |
| fees_seller_usd | Float | The absolute amount of fees the seller has gained from this transaction expressed in US dollars (USD) | Original |
| fees_opensea_usd | Float | The absolute amount of fees OpenSea has gained from this transaction expressed in US dollars (USD) | Original |
| payout_collection_address | String | The wallet address where seller fees are deposited | Anonymized |
| tx_timestamp | String | Timestamp of the transaction expressed in yyyy-mm-ddTHH:MM:SS | Original |
| price | Float | The price of the transaction expressed in token | Original |
| gain | Float | The gain after fees (i.e., gain = price - fees_opensea * price - fees_seller * price) | Original |
| usd_price | Float | The price of the transaction expressed in US dollars (USD) | Original |
| usd_gain | Float | The difference between the price and the fees expressed in US dollars (USD) | Original |
| token | Categorical | The token type used to pay the transaction | Original |
| to_eth | Float | The conversion rate to convert tokens into Ethereum at the current timestamp, such that eth = price * to_eth | Original |
| to_usd | Float | The conversion rate to convert tokens into US dollars (USD) at the current timestamp, such that usd = price * to_usd | Original |
| from_account | String | The address that sends the payment (i.e., winner/buyer) | Anonymized |
| to_account | String | The address that receives the payment (it often corresponds to the contract linked to the asset) | Anonymized |
| seller_account | String | The address of the NFT seller | Anonymized |
| winner_account | String | The address of the NFT buyer | Anonymized |
| contract_address | String | The contract address on the blockchain | Anonymized |
| created_date | Timestamp | The date of creation of the contract | Original |
| chain | Categorical | The blockchain where the transaction occurs | Original |
| token_type | Categorical | The schema of the token, i.e., ERC721 or ERC1155 | Original |
| asset_contract_type | Categorical | The asset typology, i.e., non-fungible or semi-fungible | Original |
| asset_type | Categorical | Whether the asset was involved in a simple or bundle transaction | Original |

## C   Edge Homophily Trends in Datasets

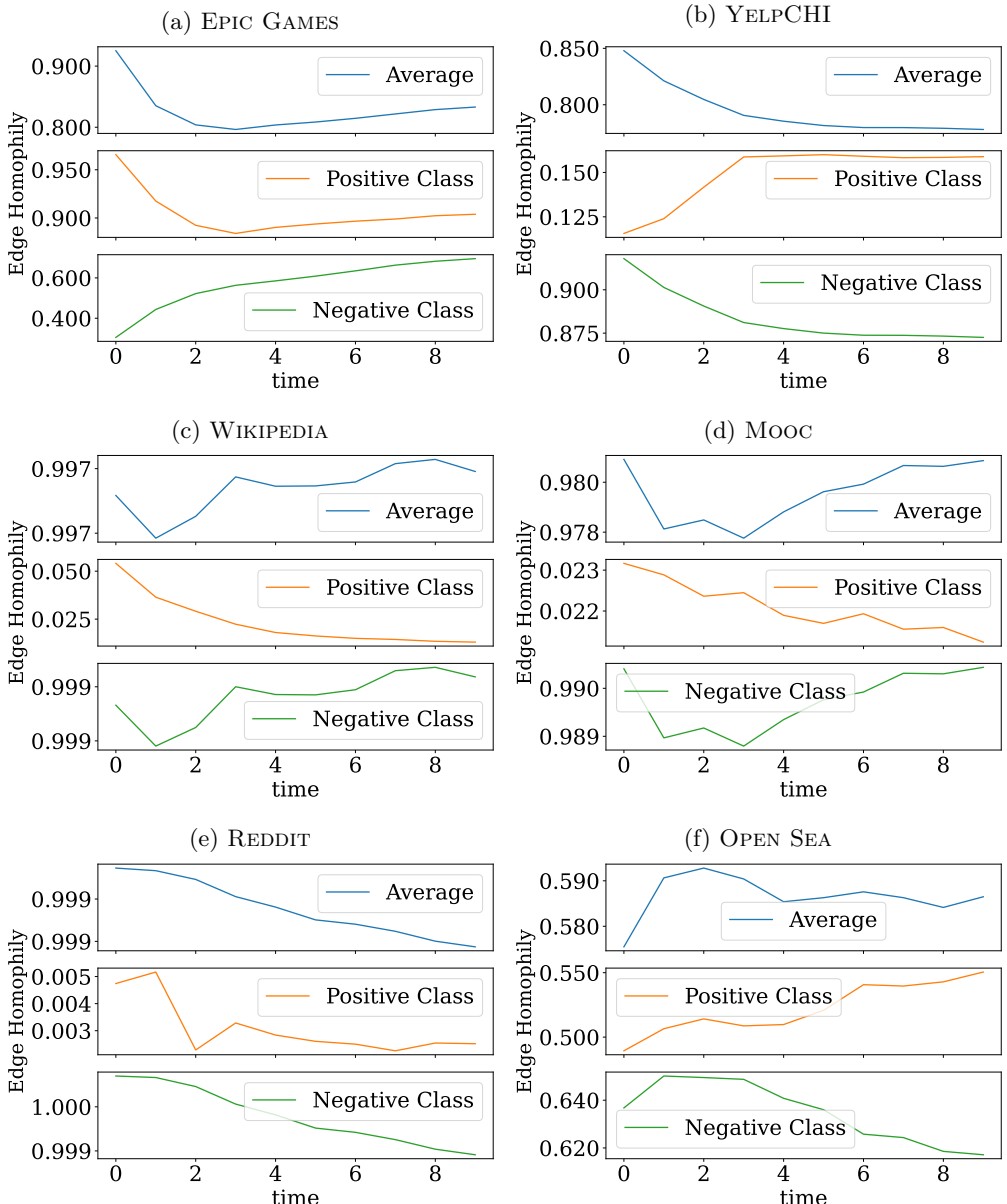

Figure 6: Edge homophily (measured using the metric Equation 6) trends as a function of time (with time measured in batches). In the case of WIKIPEDIA, MOOC, and REDDIT, the overall edge homophily reaches as high as 99%, whereas the homophily among positive class edges falls below 5%, a phenomenon attributed to significant class imbalance. Across these three platforms, there is a noticeable trend of decreasing positive edge homophily over time, a pattern that emerges as banned users or students who withdraw from courses exit the network. Conversely, on YELPCHI, there's an observable increase in positive class homophily and a decrease in negative class homophily. This trend could stem from spam attacks on businesses, typically carried out by similar groups of reviewers whose numbers grow over time. A similar pattern is detected on Open Sea, suggesting an increasing ratio of sellers who profit over time. Nonetheless, those who profited in the system's early stages remain active participants.

## D   Parameter Sensitivity Analysis

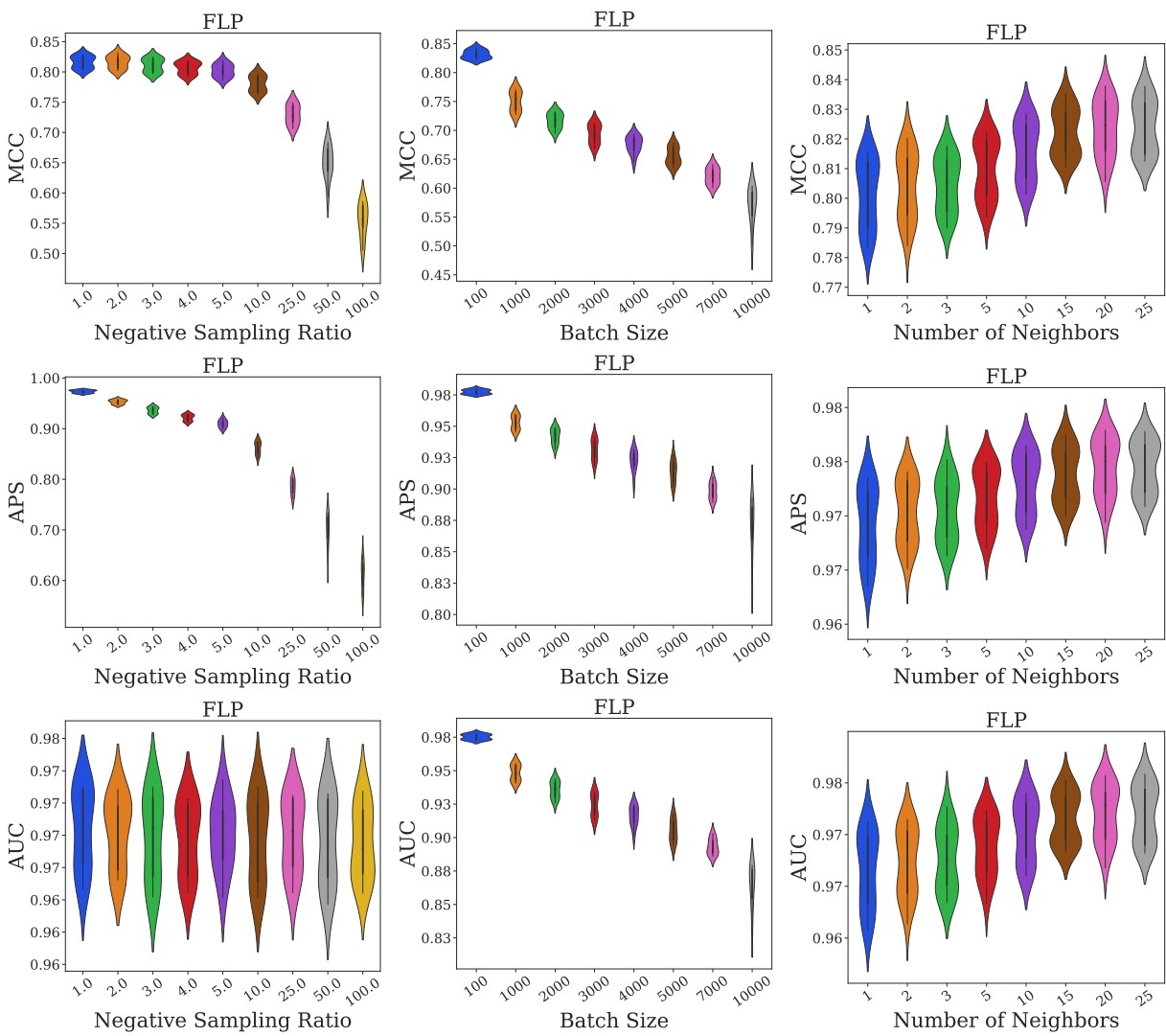

Figure 7: Parameter Sensitivity of FLP - Wikipedia

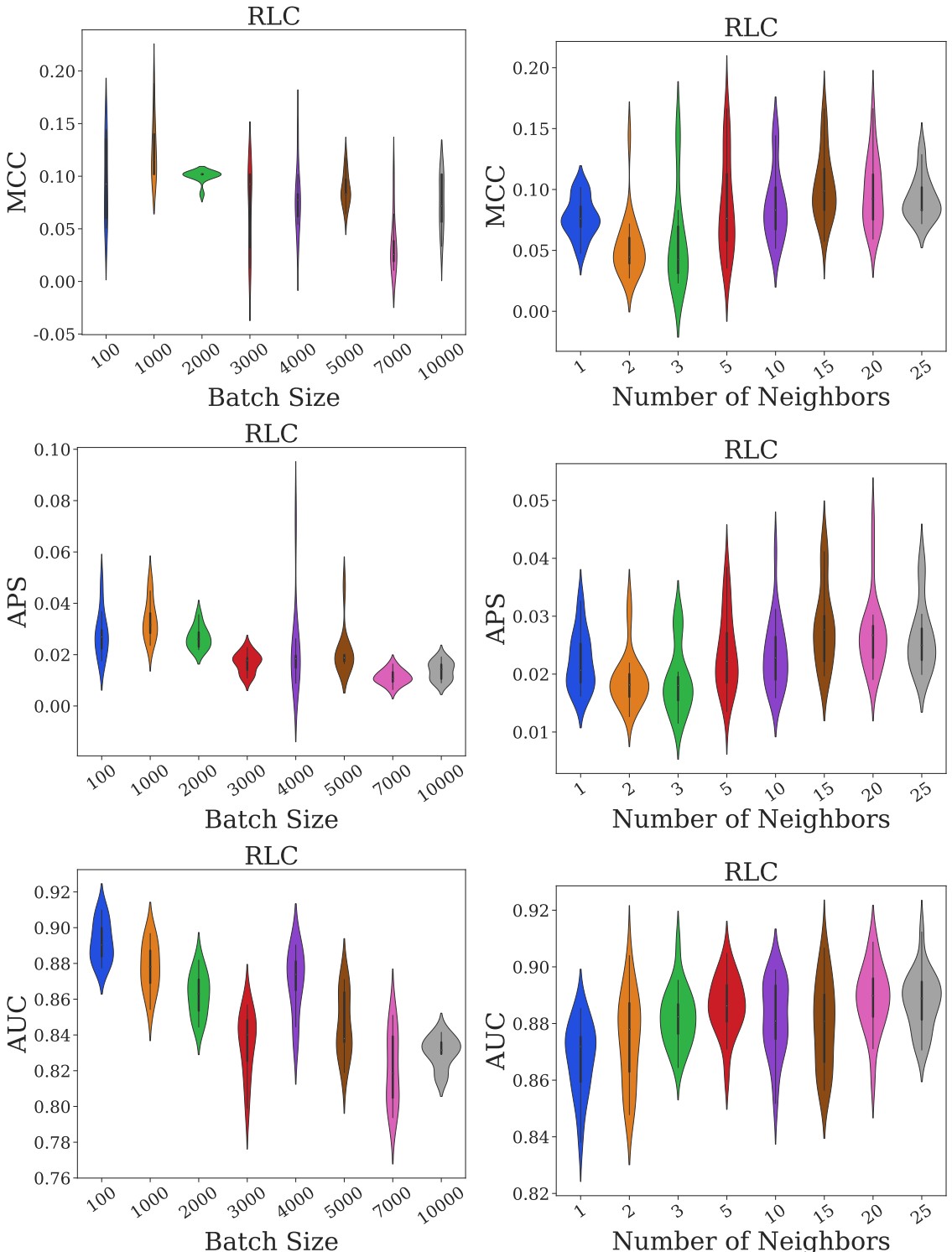

Figure 8: Parameter Sensitivity of RLC - WIKIPEDIA

## E   TGN Modifications on RLC

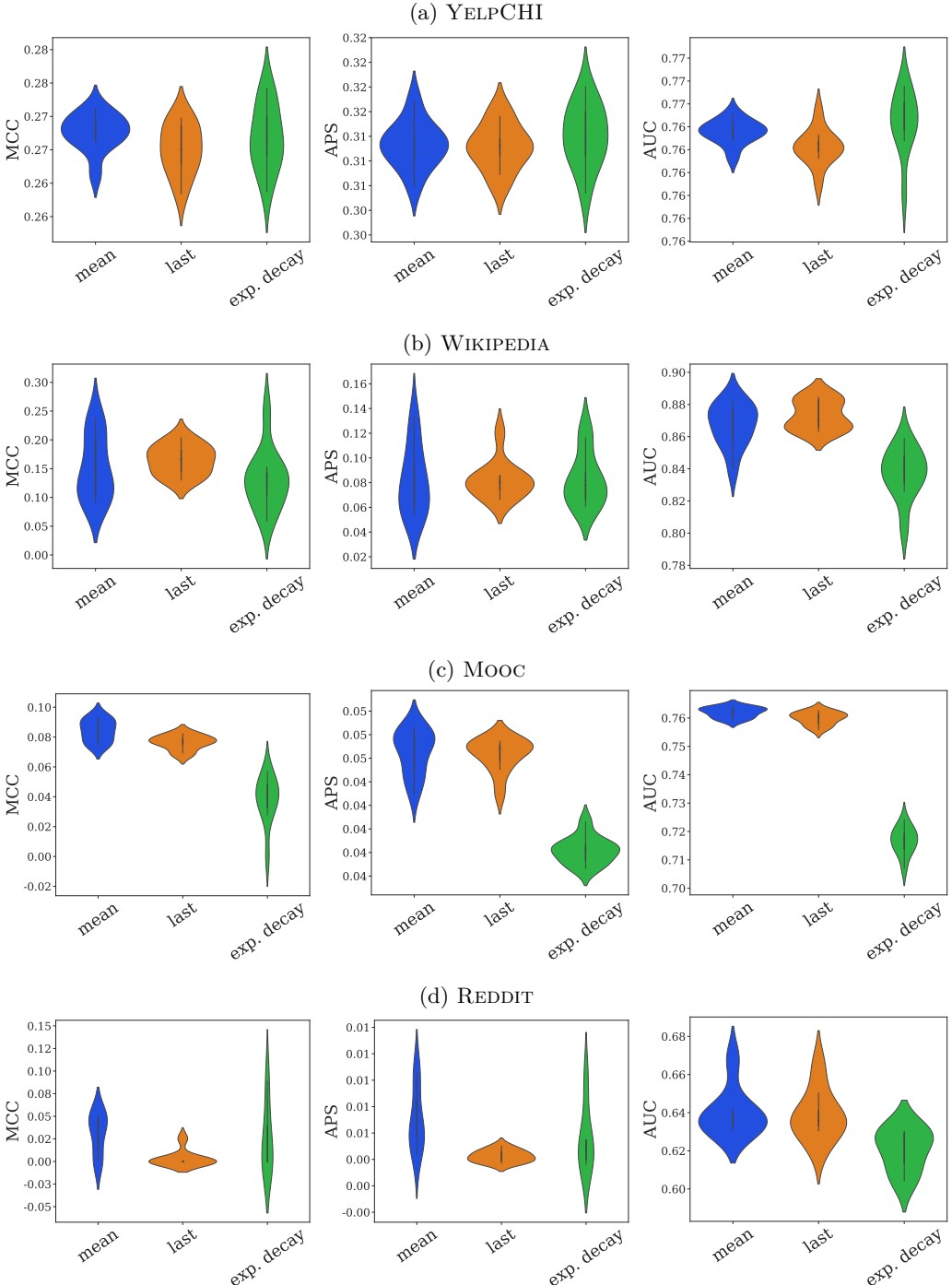

Figure 9: Aggragator Versions

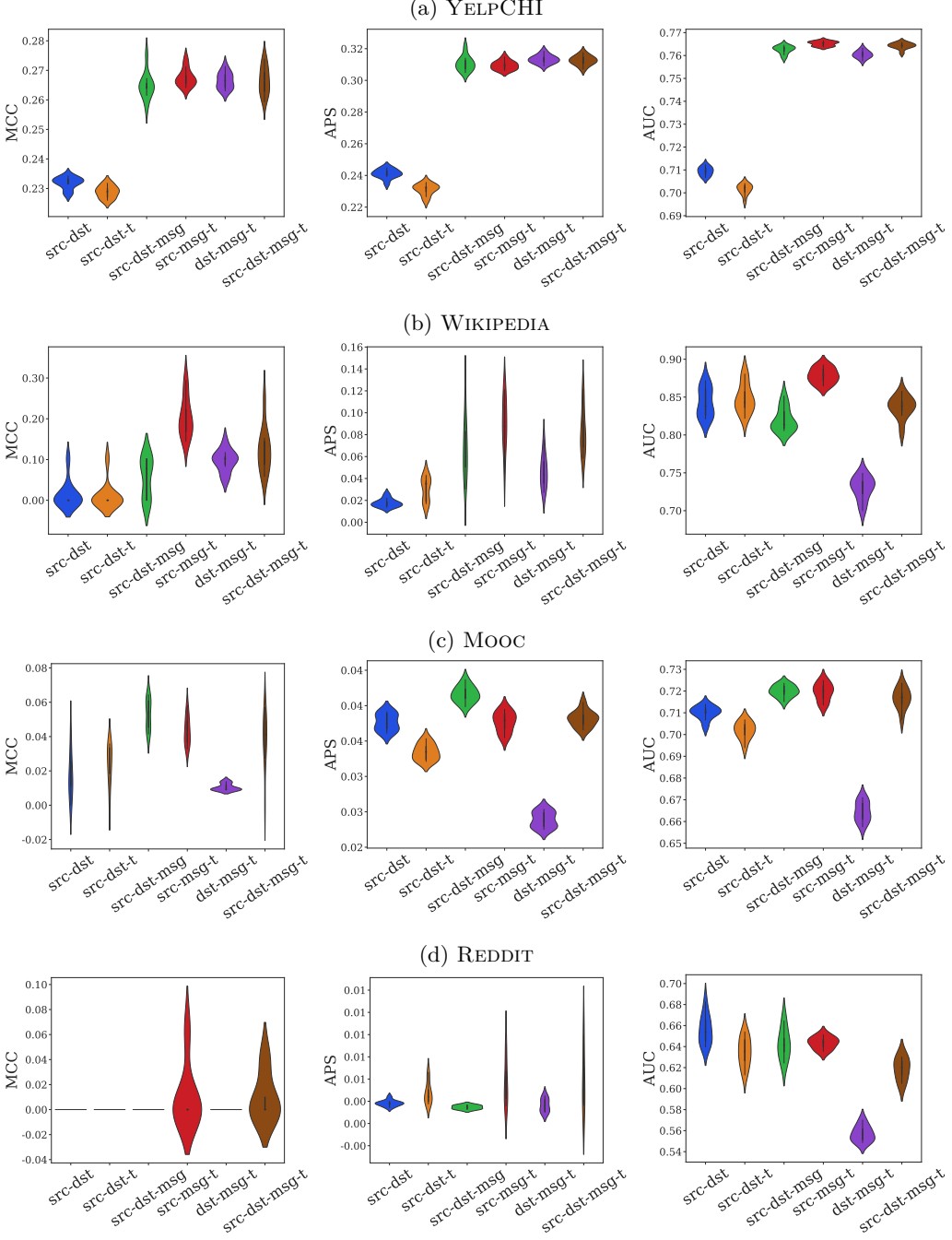

Figure 10: Readout Versions

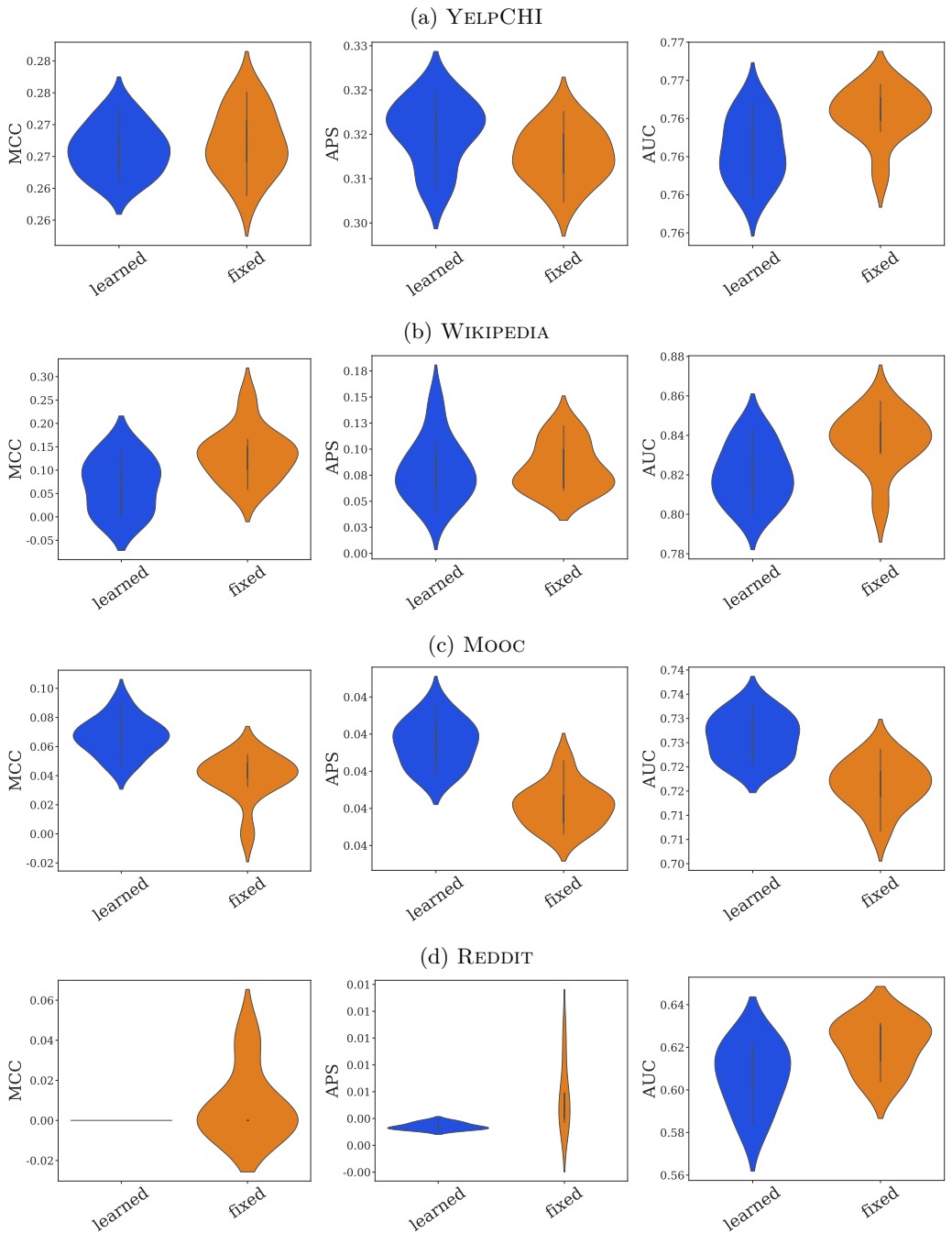

Figure 11: Time Encoding Versions

## F   Readout Variations on RLC

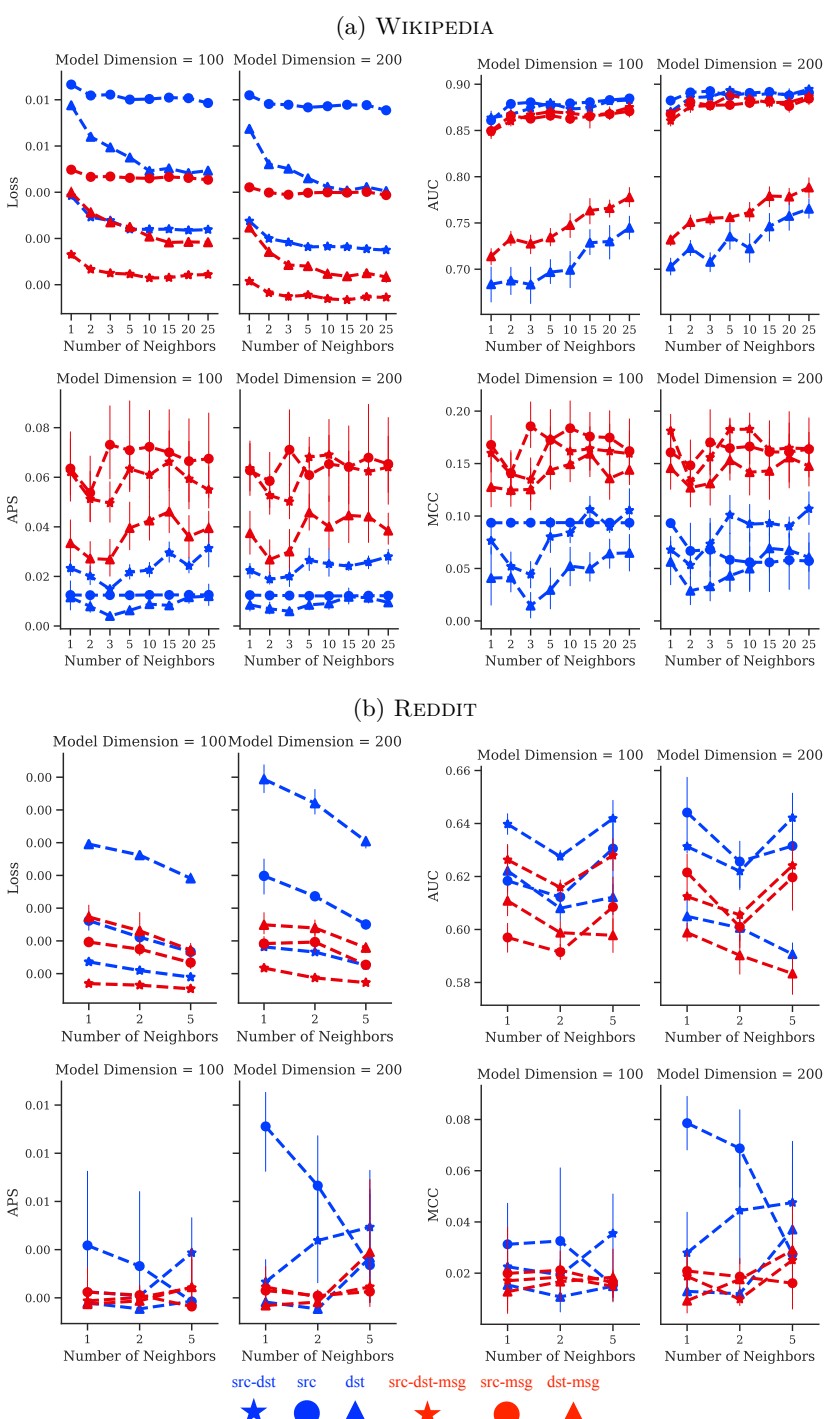

Figure 12: The performance on RLC using different variations of readout layer

