# OpenReview forum: "Recent Link Classification on Temporal Graphs Using Graph Profiler"
_TMLR — Accepted by TMLR_

### Review · Reviewer_qa8i · 2024-02-03

**Summary Of Contributions:**

The paper formalizes the Recent Link Classification (RLC) task within temporal graphs as an established benchmark downstream activity in Temporal Graph Learning (TGL), accompanied by the provision of pertinent datasets. It elucidates the distinctions between RLC and other related tasks and presents a novel architecture named Graph Profiler. The model is specifically devised to acquire entity profiles or time-aggregated representations conducive to the RLC task. It not only expands the scope of TGL but also introduces a new evaluation metric, the Matthews Correlation Coefficient. The experimental study demonstrates the superiority of the proposed model over state-of-the-art methods in TGL.

**Audience:**

Yes

**Claims And Evidence:**

Yes

**Requested Changes:**

Please refer to Weaknesses.

**Strengths And Weaknesses:**

Strengths
1. The paper formalizes a new task of Recent Link Classification and proposes a new architecture of Graph Profiler, with the design of each component clearly illustrated.
2. The paper explores the usage of Matthews Correlation Coefficient (MCC) as a novel evaluation metric on the RLC task. MCC solves the problem of skewed measures of quality when applied to imbalanced datasets.
3. The paper modifies existing TGL datasets to suit the RLC task, along with the introduction of new RLC datasets (e.g., Epic Games and Open Sea). This enriches the set of available datasets for this research problem.
4. The experimental results demonstrate that the proposed Graph Profiler consistently attains an enhanced MCC across the majority of relevant cases.

Weaknesses
1. Some figures are not cited in the main text, e.g., Figure 1 and Figure 3. Please cite them in proper places of the paper.
2. The markers in Figure 4 are hard to distinguish at the first glance.

---

### Review · Reviewer_YZjS · 2024-02-13

**Summary Of Contributions:**

The authors propose the task of recent link classification (RLC) as an alternative to the typical future link prediction (FLP) and dynamic node classification tasks used to evaluate temporal graph learning (TGL) methods. In the RLC task, we are given four pieces of information about a recently-observed edge: source node, destination node, edge timestamp, and edge features. Given these four pieces of information, the objective is to predict the class label of the edge, which is not observed until later in time.

The main contributions I observe are as follows:
- Formulation of a new task, recent link classification, for evaluating temporal graph learning methods.
- Development of the Graph Profiler architecture for the RLC task on temporal graphs with edge attributes and classes.
- Experimental evaluation on differences between RLC and FLP tasks and their implications towards designing and training temporal graph neural networks (TGNNs).

**Audience:**

Yes

**Broader Impact Concerns:**

No broader impact statement is present. I suggest that the authors add one and think about potential misuses of edge classification methods, if successful.

**Claims And Evidence:**

No

**Requested Changes:**

Major issues:
- Perform a literature search on related tasks such as link sign prediction and edge classification. These should be, at the very least, cited and discussed. It would be even stronger to have a comparison against a representative method from this category.

Minor issues:
- Better motivate the reasoning behind the asymmetry for the treatment of the source and destination nodes. Why such a complex profile encoder for the source node while having just a look-up table embedding for the destination node?
- Table 1 notation inconsistency: caption uses $\tilde{\mathcal{H}}_e^b$ while the table uses $\tilde{\mathcal{H}}_e$.

**Strengths And Weaknesses:**

Strengths:
- Proposes a new and relevant problem setting for temporal graph learning rather than continuing to make incremental improvements to existing tasks such as link prediction.
- Proposes a new measure of edge homophily to capture the importance of graph information for the RLC task.
- Empirical findings about differences in training TGNNs for RLC compared to link prediction tasks may be useful to practitioners.

Weaknesses:
- The problem setting is not as novel as claimed. The RLC task is closely related to the task of link sign prediction, which aims to predict the sign of an edge given the nodes that formed the edge, as well as edge classification. There is some work also on link sign prediction for temporal graphs. See references below for a few examples of work on link sign prediction. These should be discussed, if not compared to. This is a major weakness.
- The proposed name "recent link classification" is somewhat limiting. I'm not sure why "recent" is there, as the approach the authors consider applies to general link classification problems. The more general term "edge classification" may be the best term for this problem.
- The authors do not explain why there is an asymmetry in how the source and destination nodes are treated. There is a complex profile encoder for the source node, while the destination node is basically a standard embedding representation for a look-up table.

References on link sign prediction and edge classification:
- Jushuo Chen, Feifei Dai, Xiaoyan Gu, Haihui Fan, Jiang Zhou, Bo Li, and Weiping Wang. 2023. Learning Pair-Centric Representation for Link Sign Prediction with Subgraph. In Proceedings of the 32nd ACM International Conference on Information and Knowledge Management (CIKM '23). Association for Computing Machinery, New York, NY, USA, 256–265. https://doi.org/10.1145/3583780.3614951
- C. Aggarwal, G. He and P. Zhao, "Edge classification in networks," 2016 IEEE 32nd International Conference on Data Engineering (ICDE), Helsinki, Finland, 2016, pp. 1038-1049, doi: 10.1109/ICDE.2016.7498311. keywords: {Social network services;Gold;Context;Data mining;Electronic mail;Algorithm design and analysis;Probabilistic logic},
- Song, D., & Meyer, D. A. (2015). Link sign prediction and ranking in signed directed social networks. Social network analysis and mining, 5, 1-14.
- Dang, Q. V., & Ignat, C. L. (2018, October). Link-sign prediction in dynamic signed directed networks. In 2018 IEEE 4th International Conference on Collaboration and Internet Computing (CIC) (pp. 36-45). IEEE.

---

> ### Author Response · Authors · 2024-04-03
> **Revision changes relevant to the review**
>
> Thank you once again for your insightful feedback. In response to the reviews received, we have updated our manuscript with a revision that includes: (1) an ablation study on the various components of our proposed model, Graph Profiler; (2) a review of three distinct research areas that hold relevance to the Recent Link Classification challenge, alongside minor adjustments for clarity and correction. Currently, we are assessing the performance of additional baseline methods as recommended by one of the reviewers, which has extended the timeframe for submitting the revised version of our paper. We seize this moment to outline how the modifications in our revision tackle the issues you highlighted:
>
> - We have added a paragraph reviewing  work related to link sign prediction literature in Section 3.
> - In section 5.2 we clarified the design choice on discriminating between source and destination embedding calculation methods by adding these sentences: In the majority of bipartite graphs examined, there is a significantly larger number of source nodes compared to destination nodes. This discrepancy allows for the practicality of learning a unique embedding vector for each destination node.

---

### Review · Reviewer_jUR9 · 2024-03-22

**Summary Of Contributions:**

This paper introduces and formalizes a new task of recent link classification on temporal graphs and provides new benchmark datasets for this downstream task. To evaluate the models' performance, the authors in addition to AUC and AP further suggest using the Matthews Correlation Coefficient due to its robustness to imbalanced datasets. The authors further propose Graph Profiler, a new architecture that using its four modules (Profile Encoder, Destination Embedding, Message Encoder, Time Encoder) can encode the previous events’ class information on source and destination nodes.

**Audience:**

Yes

**Claims And Evidence:**

No

**Requested Changes:**

Please see weaknesses.

**Strengths And Weaknesses:**

`Strengths:`
---
1. Formulation of the new task of recent link classification is very interesting and important specifically in industry. Also, the provided benchmark datasets for this task can provide a great opportunity for future work.
2. The authors design Graph Profiler, which is very intuitive and simple, yet effective for this task.
3. The paper is overall well-presented and its several illustrative figures help a lot in understanding of introduced concepts.




`Weaknesses:`
---
1. What is the relationship between temporal anomaly detection and recent link binary classification? To my understanding, temporal anomaly detection methods [1, 2] can also be applied in this setting as they classify the recent links into 1 (anomaly) and 0 (normal). Also, there is a close connection with temporal multiplex networks (graphs with multiple types of edges) as the general recent link prediction can be seen as future link prediction in multiplex networks (at least with some modification they can be applied in this setting). To classify recent links, anomaly detection in temporal multiplex networks can also (see for example [3]) be applied in this setting. If this is the case, comparison and discussion of these methods are necessary.
2. It would be great if the authors could also discuss the relevance of recent link prediction with the task of dynamic link property prediction proposed in the Temporal Graph Benchmark paper [4]. Can existing methods on this task be simply modified for the recent link classification task?
3. The ablation study does not support the importance of the architecture design of Graph Profiler. It would be great if the authors could also show the impact of removing each of the modules. Is the choice of time encoding arbitrary? How learnable time encoding can affect performance?
4. Based on the reported results by [4], the state-of-the-art models can vary based on the datasets and their properties. The current choice of baselines needs improvement. It would be better if the authors could use more state-of-the-art models as baselines like DyGFormer [5] and CAWN [6].


`References:`
---
[1] AddGraph: Anomaly Detection in Dynamic Graph Using Attention-based Temporal GCN. Zheng et al. 2019.
[2] Netwalk: A flexible deep embedding approach for anomaly detection in dynamic networks. Yu et al. 2018.
[3] Anomaly Detection in Multiplex Dynamic Networks: from Blockchain Security to Brain Disease Prediction. Behrouz et al. 2022.
[4] Temporal Graph Benchmark for Machine Learning on Temporal Graphs. Huang et al. 2023.
[5] Towards Better Dynamic Graph Learning: New Architecture and Unified Library. Yu et al. 2023.
[6] Inductive Representation Learning in Temporal Networks via Causal Anonymous Walks. Wang et al. 2021.

---

> ### Author Response · Authors · 2024-04-03
> **Revision changes relevant to the review**
>
> Thank you once again for your insightful feedback. In response to the reviews received, we have updated our manuscript with a revision that includes: (1) an ablation study on the various components of our proposed model, Graph Profiler; (2) a review of three distinct research areas that hold relevance to the Recent Link Classification challenge, alongside minor adjustments for clarity and correction. Currently, we are assessing the performance of additional baseline methods as recommended by one of the reviewers, which has extended the timeframe for submitting the revised version of our paper. We seize this moment to outline how the modifications in our revision tackle the issues you highlighted:
>
> - We have added paragraphs reviewing  work related to 1- dynamic link property prediction and 2- dynamic graph anomaly detection anomaly detection literature in Section 3.
> - We performed an ablation study removing the components of our proposed model, Graph Profiler:
>
> This experiment involved systematically removing specific module from the model and comparing the outcomes with those of the intact version. For a more in-depth comparative analysis, we utilized datasets Yelpchi and Epic Games. These were chosen for their comprehensive content and manageable sizes, facilitating the intensive computations required for a thorough statistical evaluation. The findings are presented in table below where each row reports the results with the configuration without the indicated module. Results show that while each component plays a critical role, their significance varies across contexts. For instance, the profile encoder proves to be exceptionally valuable for the Epic Games dataset, whereas encoding features of recent interactions has a more pronounced effect on the Yelpchi dataset. This suggests that, in the context of the Epic Games platform, the profile information of reviewers holds greater predictive power for identifying top critiques than the content of their reviews. Conversely, for Yelpchi, the content of reviews is more crucial in detecting spam.
> |  | Epic Games | YelpCHI | Epic Games | YelpCHI | Epic Games | YelpCHI |
> |---|:---:|:---:|:---:|:---:|:---:|:---:|
> |  | MCC |  | APS |  | AUC |  |
> | Graph Profiler | 0.9355 | 0.3274 | 0.9988 | 0.4059 | 0.9974 | 0.8058 |
> | Profile Encoder | 0.0000 | 0.2849 | 0.6097 | 0.3316 | 0.5000 | 0.7725 |
> | Learning Destination Embeddings | 0.7695 | 0.3071 | 0.9575 | 0.3892 | 0.9054 | 0.8000 |
> | Recent Event Message Encoder | 0.6311 | 0.1926 | 0.8940 | 0.2215 | 0.8593 | 0.6568 |
> | Recent Event Time Encoder | 0.8082 | 0.3072 | 0.9766 | 0.3970 | 0.9590 | 0.7986 |
> The results of this study is reported in Appendix G.
> - We are currently working on adding more baselines to our comparisons as you have suggested. This  is taking some time, since it requires modifications specific to RLC in addition to data preprocessing to match. We aim to finalize by Friday and upload the revision with that.

---

> > ### Author Response · Authors · 2024-04-13
> > **Added another baseline's results**
> >
> > Thank you once more for highlighting the potential baseline models for us. We have successfully replicated the results of CAWN [6] and updated our paper accordingly. Notably, CAWN demonstrated strong performance on one of our datasets, Open Sea, and performed comparably to our own method, Graph Profiler, on this dataset. However, as a multi-head attention-based architecture, CAWN required significantly more wall clock time to tune and conduct experiments with the same number of random seeds compared to both Graph Mixer and Graph Profiler.

---

### Author Response · Authors · 2024-04-03
**Revision-1**

We would like to express our gratitude for the insightful feedback provided by the reviewers. In response to their valuable comments, we have implemented modifications to our paper, outlined as follows:

- A comprehensive review of TGL literature has been conducted, resulting in the expansion of Section 2.
- We have explored the relevance of three subfields within GRL literature to our benchmark task, providing a succinct review of each in Section 3.
- An ablation study has been carried out on the individual components of our proposed method, which is detailed in Section 5.5.

Alongside these major revisions, we have also made a number of minor adjustments throughout the text, including updates to figure references and appendices.

Following a suggestion from one of our reviewers, we are in the process of incorporating additional state-of-the-art methods for comparison. This enhancement is underway and will be included in a forthcoming revision.

It is important to note that these amendments have increased the overall content of our manuscript, resulting in a transition from 12 to 13 main content pages. We would like to check with the AE to ensure that this is acceptable. If not, we will perform further revisions to move some of our discussion to the appendix.

Best regards
The Authors

---

### Author Response · Authors · 2024-04-17
**Rebuttal Revisions**

Dear Reviewers,

In general, thank you very much for your thorough reviews and feedbacks. In the light of your comments, we have revised our paper. The main changes are as follows:
- A comprehensive review of TGL literature has been conducted, resulting in the expansion of Section 2.
- We have explored the relevance of three subfields within GRL literature to our benchmark task, providing a succinct review of each in Section 3.
- An ablation study has been carried out on the individual components of our proposed method, which is detailed in Section 5.5.
- We adapted the implementation of another baseline to the proposed problem and added to our model comparison.

Alongside these revisions, we have also made a number of minor adjustments throughout the text, including updates to figure references and appendices.

As the rebuttal period concludes, we believe that we have effectively addressed all your inquiries and questions. Should you have any additional questions or require further clarification, please feel free to reach out. We appreciate your time and consideration.

Best regards,

The Authors

---

### Decision · Action_Editor_iZju · 2024-05-05

**Recommendation:** Accept as is

**Comment:**

All reviewers agree that the paper is ready for publication with some minor changes.

**Audience:**

The paper would be of interest to a subset of the TMLR community who work on link prediction.

**Claims And Evidence:**

The paper introduces a new task of recent link classification and develops benchmarks and solutions on this task. The paper conducts solid work and is well-written. After the rebuttal, all reviewers agree that the paper is ready for publication with some minor changes. The authors are suggested to make the following changes:
1. Adding more baselines utilizing existing edge classification or link sign classification techniques.
2. Proofreading. E.g. addressing the inconsistency in the Table 1 caption and table header.